# A truncating mutation in the autophagy gene *UVRAG* drives inflammation and tumorigenesis in mice

Christine Quach [1,10], Ying Song[1,10], Hongrui Guo[1,2,10], Shun Li[1,3], Hadi Maazi [1], Marshall Fung[1], Nathaniel Sands[1], Douglas O'Connell [1], Sara Restrepo-Vassalli[4], Billy Chai[1], Dali Nemecio [1], Vasu Punj[5], Omid Akbari[1], Gregory E. Idos[6], Shannon M. Mumenthaler [7], Nancy Wu[8], Sue Ellen Martin [9], Ashley Hagiya[9], James Hicks [4], Hengmin Cui[2] & Chengyu Liang[1]*

Aberrant autophagy is a major risk factor for inflammatory diseases and cancer. However, the genetic basis and underlying mechanisms are less established. UVRAG is a tumor suppressor candidate involved in autophagy, which is truncated in cancers by a frameshift (FS) mutation and expressed as a shortened UVRAG[FS]. To investigate the role of UVRAG[FS] in vivo, we generated mutant mice that inducibly express UVRAG[FS] (*iUVRAG[FS]*). These mice are normal in basal autophagy but deficient in starvation- and LPS-induced autophagy by disruption of the UVRAG-autophagy complex. *iUVRAG[FS]* mice display increased inflammatory response in sepsis, intestinal colitis, and colitis-associated cancer development through NLRP3-inflammasome hyperactivation. Moreover, *iUVRAG[FS]* mice show enhanced spontaneous tumorigenesis related to age-related autophagy suppression, resultant β-catenin stabilization, and centrosome amplification. Thus, UVRAG is a crucial autophagy regulator in vivo, and autophagy promotion may help prevent/treat inflammatory disease and cancer in susceptible individuals.

[1] Department of Molecular Microbiology and Immunology, Keck School of Medicine, University of Southern California, Los Angeles, CA 90033, USA. [2] College of Veterinary Medicine, Sichuan Agriculture University, Chengdu 611130, China. [3] Key Laboratory of Biorheological Science and Technology, Ministry of Education, College of Bioengineering, Chongqing University, Chongqing 400044, China. [4] USC Michelson Center for Convergent Bioscience, Bridge Institute, University of Southern California, Los Angeles, CA 90089, USA. [5] Department of Medicine, University of Southern California, Los Angeles, CA 90033, USA. [6] Norris Comprehensive Cancer Center, University of Southern California, Los Angeles, CA 90089, USA. [7] Lawrence J. Ellison Institute for Transformative Medicine, University of Southern California, Los Angeles, CA 90033, USA. [8] Norris Comprehensive Cancer Center Transgenic/Knockout Rodent Core Facility, University of Southern California, Los Angeles, CA 90089, USA. [9] Department of Pathology, Keck School of Medicine, University of Southern California, Los Angeles, CA 90033, USA. [10] These authors contributed equally: Christine Quach, Ying Song, Hongrui Guo. *email: chengyu.liang@med.usc.edu

utophagy, an evolutionarily conserved catabolic process, is responsible for regulated degradation of intracellular components to maintain cellular homeostasis and adapt to environmental cues[1]. During starvation, increased levels of autophagy allow cells to breakdown cytoplasmic materials and release essential metabolites as an energy source[2]. Recent studies[3] further link autophagy to lipid metabolism, whereby starvation-induced autophagy mediates lipid droplet degradation (lipophagy), which provides free fatty acids to maintain metabolic homeostasis. Furthermore, autophagy-mediated selective clearance of damaged mitochondria (mitophagy) attenuates mitochondrial stress upon infection or inflammation to prevent excessive mitochondrial ROS (mtROS) production, which otherwise perpetuates the inflammatory response, leading to immune pathology and tissue damage[4,5]. A lack of sufficient autophagy has been implicated in inflammatory pathologies and immune-dysfunction such as sepsis, Crohn's disease, and diabetes[6]. Under unstressed conditions, chronic autophagy suppression is associated with decreased lifespan in animal models, such as age-related renal and cardiac deterioration and spontaneous tumorigenesis[7]. However, the precise mechanisms that tie autophagy activity to age-related diseases remain incompletely understood.

The exploding wave of autophagy investigation in the past decade has led to the discovery of a plethora of autophagy-related genes that govern its induction and execution. The ultraviolet (UV) irradiation resistance-associated gene (UVRAG) is an autophagy-related factor that forms a complex with Beclin1 and the lipid kinase Vps34 to promote autophagic membrane remodeling[8]. Later, it was discovered that different UVRAG complexes exist and modulate the tightly regulated autophagosome progression to lysosomal degradation, as well as other membrane trafficking events that either intersect or converge with the autophagy pathway[9–12]. Overexpression of UVRAG activates autophagy and suppresses tumor growth, whereas silencing of UVRAG causes failure of autophagy and uncontrolled cell proliferation[8,10]. On the other hand, UVRAG also exerts autophagy-independent functions in DNA repair and organelle integrity[13–15]. Notably, a frameshift (FS) mutation in UVRAG, which results in the expression of a truncated UVRAG (referred to as UVRAG^FS), is linked to susceptibility to different cancer types[16]. In addition to losing the tumor suppressing properties of wild-type (WT) UVRAG, the truncated UVRAG^FS overrides co-existing WT protein and acts as a dominant-negative mutant and autophagy suppressor in cell-based assays[16]. However, some studies and models have shown that deletion of yeast Vps38p and *Arabidopsis* Vps38, which are considered as orthologs of mammalian UVRAG, impairs vacuolar protein sorting but has minimal effect on autophagy[17,18], raising the concern whether UVRAG regulates autophagy in mammals in vivo. Unfortunately, there is no genetic evidence in mice showing a role of UVRAG in autophagy largely because genetic inactivation of UVRAG results in early embryonic lethality[19]. Hence, definitive evidence that the autophagic response succumbs to impaired UVRAG function and its impact on tissue homeostasis and disease propensity is lacking.

Herein, we generated a doxycycline (Dox)-inducible mouse model that does not affect basal autophagy but is significantly impaired in starvation- and Toll-like receptor (TLR)-induced autophagy by expression of the cancer-derived UVRAG^FS (referred to as iUVRAG^FS). Mice expressing UVRAG^FS display an enhanced inflammatory response through hyperactivated NLRP3 and IL-1β secretion primed by mitochondrial signaling. Using iUVRAG^FS mice, we furthermore demonstrate that long-term UVRAG^FS expression leads to centrosome amplification, age-related autophagy suppression, and resultant β-catenin oncoprotein stabilization, leading to increased susceptibility to

spontaneous tumorigenesis in multiple organs. To the best of our knowledge, our results provide the first genetic evidence connecting UVRAG suppression to autophagy regulation, inflammatory disorders, and cancer predisposition.

## Results

**UVRAG^FS inhibits starvation-induced autophagy in vivo.** To study the role of UVRAG^FS in a temporal-specific manner, we generated a conditional Flag-tagged UVRAG^FS-luciferase transgene under the control of a Dox-responsive element (designated TRE-UVRAG^FS) (Supplementary Fig. 1a). These mice were crossed to ROSA26-rtTA2-M2 mice[20] to enable Dox-inducible expression of human UVRAG^FS in a tightly regulated fashion (Supplementary Fig. 1a). This double transgenic strain is referred to as iUVRAG^FS. Dox treatment induced UVRAG^FS expression at the mRNA and protein levels without affecting endogenous UVRAG expression (Supplementary Fig. 1b, c). No UVRAG^FS transgene expression was detected in Dox-treated WT littermate control (Supplementary Fig. 1b, c). Luciferase expression was not detected in untreated mice but strongly correlated with UVRAG^FS expression, providing a visual biomarker for UVRAG^FS expression (Supplementary Fig. 1d–f). The transgene expression was reversible, showing loss of luciferase and UVRAG^FS expression 4 days following Dox withdrawal in different organs (Supplementary Fig. 1d–h). Dox-treated iUVRAG^FS mice were of normal size and weight, and displayed normal histology in major organs (Supplementary Fig. 1h, i).

To explore the impact of UVRAG^FS on autophagy in vivo, we bred WT control or iUVRAG^FS mice to GFP-tagged LC3 transgenic mice that express a fluorescent marker of autophagosomes[21], on C57BL/6J background, and analyzed the tissues of resultant compound mice after starvation (Fig. 1a). In skeletal muscle, liver, heart, and colon, Dox-treated iUVRAG^FS mice had significantly decreased numbers of GFP-LC3 puncta compared to controls and to Dox-untreated mice that showed a marked increase in GFP-LC3 puncta upon starvation (Fig. 1a). We further confirmed mice with UVRAG^FS expression having suppressed starvation-induced autophagy by western blot analyses (Fig. 1b). There were decreased levels of autophagosome-associated lipidated LC3 (LC3-II)[22,23] and increased levels of the autophagy substrate p62[24] in 48 h-starved iUVRAG^FS mice on Dox, while Atg16 and Atg12 conjugation to Atg5 remained unaffected (Fig. 1b). The deficient starvation-induced autophagy in Dox-treated iUVRAG^FS mice was not associated with increased cell death (Supplementary Fig. 1j). Using electron microscopy (EM), it was also observed that the numbers of autophagic vacuoles (autophagosome and autolysosome) were comparable at baseline in liver of Dox-untreated and -treated mice but failed to increase in Dox-treated iUVRAG^FS mice following starvation (Fig. 1c). In parallel with compromised starvation-induced autophagy, Dox-treated iUVRAG^FS mice showed massive enlargement and accumulation of lipid droplets (LD) in liver when compared to controls (Fig. 1c). Increased LDs were also observed in fasting WT mice, but to a much lesser extent, and LDs were mostly surrounded by autophagic vesicles that were not found in cells with UVRAG^FS expression (Fig. 1c). The marked increase of LDs was further confirmed by Oil red O staining of liver sections in starved iUVRAG^FS mice on Dox (Supplementary Fig. 1j), supporting previous findings on the potential role of autophagy in the clearance of hepatic LDs[3]. Thus, UVRAG^FS prevents starvation-induced autophagy activation in vivo, which may disturb metabolic adaptation of cells to nutrient deprivation and therefore increase the risk of metabolic disorders.

To understand how UVRAG^FS suppresses starvation-induced autophagy in vivo, we examined WT UVRAG association with

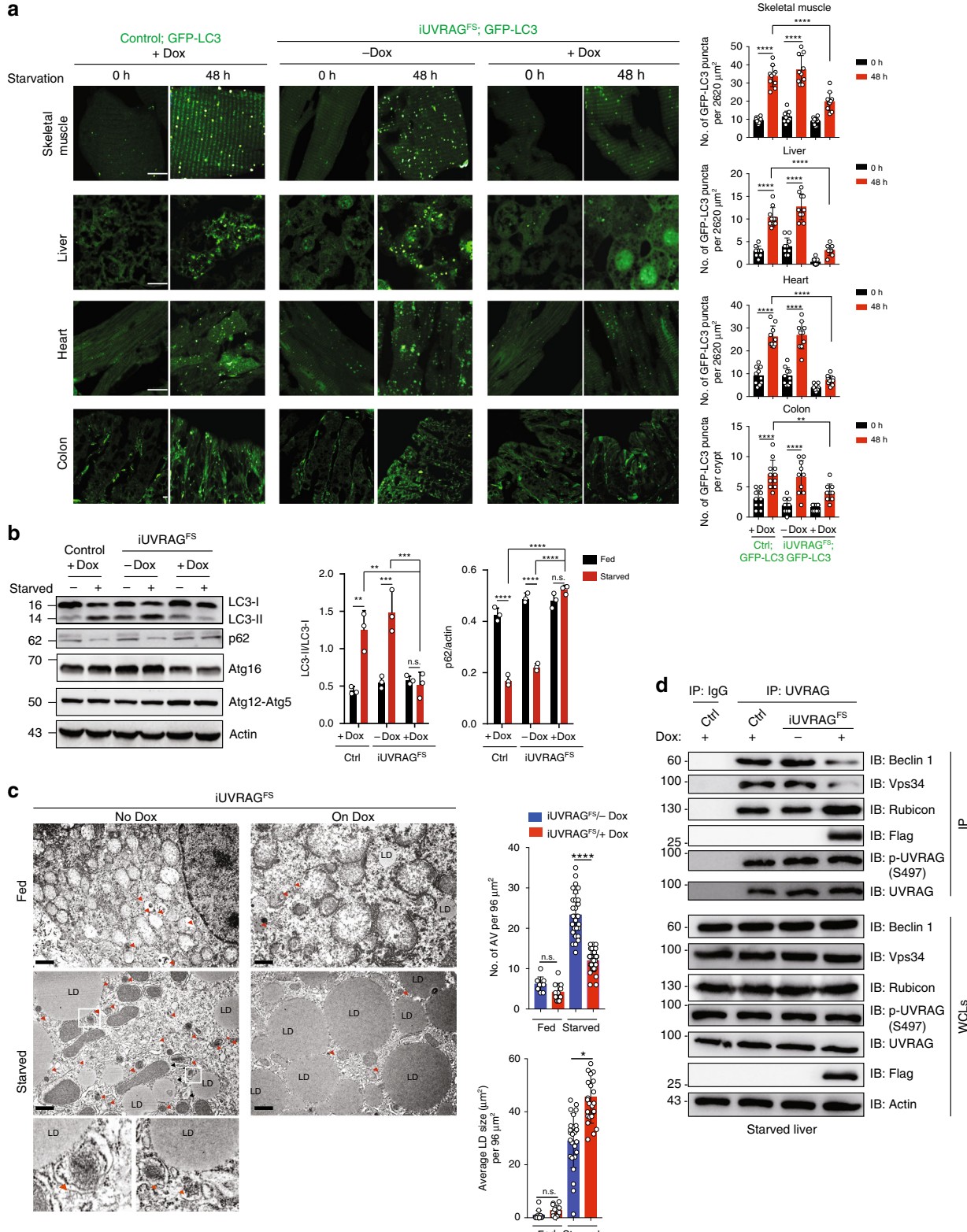

Beclin1 and Vps34, which are parts of an autophagic-specific class III phosphatidylinositol-3-OH kinase (PI3KC3) complex that has a key role in autophagosome formation[8,10]. In the 48-h-starved tissue of Dox-induced *iUVRAG^FS* mice, there was a marked reduction in UVRAG co-immunoprecipitation with Beclin1 and Vps34, concomitant with an increase in its binding to Rubicon, a negative regulator of the PI3KC3 complex[12,25] (Fig. 1d). Notably, reduced assembly of the UVRAG-containing PI3KC3 complex

was not due to alteration of UVRAG S497 phosphorylation (S498 in human UVRAG) in *iUVRAG^FS* mice (Fig. 1d), which, mediated by mTORC1, was previously shown to decrease the autophagy activity of UVRAG in vitro[26]. Rather, reduced assembly was associated with the dominant-negative effect of UVRAG^FS that bound to and sequestered UVRAG from the PI3KC3 complex, leading to autophagy suppression (Fig. 1d). These results indicate that UVRAG^FS inhibits starvation-induced autophagy by acting

**Fig. 1** UVRAG[FS] inhibits starvation-induced autophagy in vivo. **a** Representative images (left panels) and quantification (right panels) of GFP-LC3 puncta in indicated tissues from Dox-treated/untreated iUVRAG[FS] mice and control mice crossed with GFP-LC3 transgenic mice, following 48 h of starvation. n = 10 tissue sections pooled from three independent experiments. Scale bars, 10 μm. **b** Western blot analysis of LC3-I/II, p62 levels, Atg12-Atg5 conjugates, and Atg16 levels in the liver from Dox-treated/untreated iUVRAG[FS] mice and control mice following 48 h starvation. The densitometric quantification of the LC3-II/LC3-I and the p62/actin ratios under the indicated conditions are shown (right panels). **c** Representative electron microscopy (EM) images of the liver from Dox-treated/untreated iUVRAG[FS] mice in fed or starved conditions for 48 h. Note the accumulation of lipid droplets (LDs) in starved iUVRAG[FS] mice on Dox. Insets highlight LDs being surrounded by autophagic membrane structures. Arrowheads (red) denote autophagic vacuoles. The number and size of LD and the number of autophagic vacuoles (AV) in liver from indicated mice were quantified (right panels). n = 10–30 cells pooled from three independent experiments. Scale bars, 1.0 μm. **d** Co-immunoprecipitation (co-IP) of autophagy-related proteins with UVRAG in 48 h-starved liver from control and Dox-treated/untreated iUVRAG[FS] mice. IgG is a negative control. Actin serves as a loading control. IP, immunoprecipitated; WB, western blot; WCL, whole cell lysates. Data in **b–d** are from one experiment that is representative of three independent experiments. For all quantifications, data (mean ± SD) were from the indicated number of independent experiments and analyzed with two-way ANOVA. n.s., not significant; *P < 0.05; **P < 0.01; ***P < 0.001; ****, P < 0.0001. Source data are provided as a Source Data file. See Supplementary Fig. 9 for uncropped data of **b, d**.

on the UVRAG-autophagy complex, and that mammalian UVRAG, unlike its predicted orthologue of yeast Vps38p[27], is required for starvation-induced autophagy activation in vivo.

**UVRAG[FS] prevents TLR4 induction of autophagy in vivo.** Besides starvation, autophagy is induced by TLRs signaling as part of innate immunity in response to microbial components[28,29]. To evaluate the impact of UVRAG[FS] on autophagy induced by lipopolysaccharide (LPS), a major component of bacterial endotoxin that activates the TLR4 microbe sensor[30], we challenged control and UVRAG[FS]-expressing mice crossed with GFP-LC3 mice with LPS-induced septic shock. We observed that LPS stimulation increased GFP-LC3 puncta in different tissues (liver, colon, and lung) of control mice, while Dox-treated iUV-RAG[FS] mice displayed marked impairment of LPS-induced upregulation of autophagy (Fig. 2a). Analogous results were also obtained in bone marrow-derived macrophages (BMDMs) from iUVRAG[FS] mice, whereby LPS induced a bona fide increase in both numbers of autophagosomes (indicated by GFP+RFP+ puncta) and autophagic flux (indicated by GFP-RFP+ puncta) in WT mice but not in mice with UVRAG[FS] expression, as indicated by the tandem fluorescently tagged mRFP-EGFP-LC3[31] (Supplementary Fig. 2a). These results indicate that UVRAG[FS] prevents LPS-induced autophagy activation.

LPS engagement of TLR4 and resultant autophagy induction requires MyD88- and/or TRIF-dependent TRAF6 (E3-ubiquitin ligase) activation and TRAF6-mediated K63-linked ubiquitination of Beclin1 in macrophage in vitro[32,33]. We examined Beclin1 recruitment to the TLR4 complex and consequent modification by TRAF6 in LPS-treated BMDM. Co-immunoprecipitation analysis showed that LPS treatment induced interaction of Beclin1 with MyD88, TRIF, and TRAF6, which was associated with induced K63-linked Beclin1 ubiquitination in BMDM (Fig. 2b). Dox treatment decreased LPS-induced Beclin1 interaction with MyD88 (but not TRIF) and TRAF6, as well as Beclin1 K63 ubiquitination (Fig. 2b). By contrast, Beclin1 interaction with its negative regulator Bcl-2 was increased in LPS-stimulated iUVRAG[FS] BMDM as compared to control cells under the same treatment (Fig. 2b). We also noted that UVRAG[FS] was able to dissociate the UVRAG-Beclin1-PI3KC3 complex, but had minimal effect on the Atg14-Beclin1-PI3KC3 complex in LPS-stimulated iUVRAG[FS] BMDM (Fig. 2b). The suppression of the Beclin1 interactome by UVRAG[FS] was associated with reduced PI3KC3 kinase activity in LPS-treated BMDM (Fig. 2c) as illustrated by decreased punctate staining of the PI3KC3 product, phosphatidylinositol 3-phosphate (PI3P)[34]. Similar results were obtained in livers of LPS-treated iUVRAG[FS] mice (Supplementary Fig. 2b). These results indicate that UVRAG[FS] plays an inhibitory role in TLR4-induced autophagy by interfering with the dynamics of Beclin1-containing TLR4 signaling complex formation and autophagy activity (Supplementary Fig. 2c).

**Enhanced inflammatory responses by UVRAG[FS].** In parallel with impaired LPS-induced autophagy in Dox-treated iUVRAG[FS] mice, there was increased vulnerability to intraperitoneally injected LPS (Supplementary Fig. 3a). The survival of UVRAG[FS]-expressing mice in septic shock was significantly lower than that of controls (no Dox) (Supplementary Fig. 3a). Notably, Dox-treated iUVRAG[FS] mice had similar levels of serum TNF-α, IL-6, and IFN-β after LPS stimulation as control mice (Fig. 3a). Characterization of NF-κB activation in BMDM from control and UVRAG[FS]-expressing mice revealed no discernable difference in the phosphorylation and degradation of the NF-κB inhibitor IκBα[35] (Supplementary Fig. 3b) and in nuclear translocation of p65 (Supplementary Fig. 3c). Consistently, the expression of Tnf-a, Il-6, and Ifn-b mRNA in spleens was comparable between LPS-stimulated UVRAG[FS]-expressing mice and control mice (Supplementary Fig. 3d). Unlike TNF-α, whose synthesis and secretion are inflammasome-independent, production of IL-1β and IL-18, which requires inflammasome-dependent caspase-1 activation[4,36], was more abundant in the serum of Dox-induced iUVRAG[FS] mice than littermate control in sepsis (Fig. 3a). However, their mRNA synthesis was not increased accordingly (Supplementary Fig. 3d). To determine whether UVRAG[FS] plays a role in LPS-induced inflammasome activation, we stimulated control and UVRAG[FS]-expressing BMDM with LPS and the NLRP3 inflammasome activator ATP and observed more than two-fold increase in mature IL-1β production, but not pro-IL-1β levels, by Dox-induced iUVRAG[FS] macrophages compared with control cells, suggesting that enhanced IL-1β secretion is likely related to increased inflammasome activation (Fig. 3b,c). As expected, there was a notable increase in caspase-1 activation (indicated by cleaved caspase-1) in LPS + ATP-treated iUVRAG[FS] BMDM on Dox, whereas activation of the inflammasome-independent proinflammatory caspase, caspase-11[37], by LPS was similar in UVRAG[FS]-expressing BMDM relative to its activation in control cells (Fig. 3b, c). Elevated caspase-1 activation and IL-1β production correlated with increased rate of macrophage death (Supplementary Fig. 3e) and IL-1β-inducible chemokines CXCL1, CXCL2, and CCL2 expression in spleens (Supplementary Fig. 3f). Treatment of iUVRAG[FS] BMDM with Z-YVAD-FMK, a caspase-1 inhibitor, ablated enhanced IL-1β secretion by UVRAG[FS] (Fig. 3d), confirming that UVRAG[FS]-associated increased levels of IL-1β was the consequence of increased caspase-1 activation. Thus, caspase-1 overactivation contributes to excessive inflammation in Dox-treated iUVRAG[FS] mice after LPS stimulation.

Caspase-1 is activated when it is recruited and incorporated into inflammasomes, which contains a Nod-like sensor protein such as NLRP3 and the adapter molecule ASC[38]. We observed that UVRAG[FS]-expressing BMDM demonstrated a marked increase in NLRP3 and ASC co-immunoprecipitation with pro-caspase-1 than in control BMDM, suggesting increased NLRP3 inflammasome assembly upon UVRAG[FS] expression (Fig. 3b).

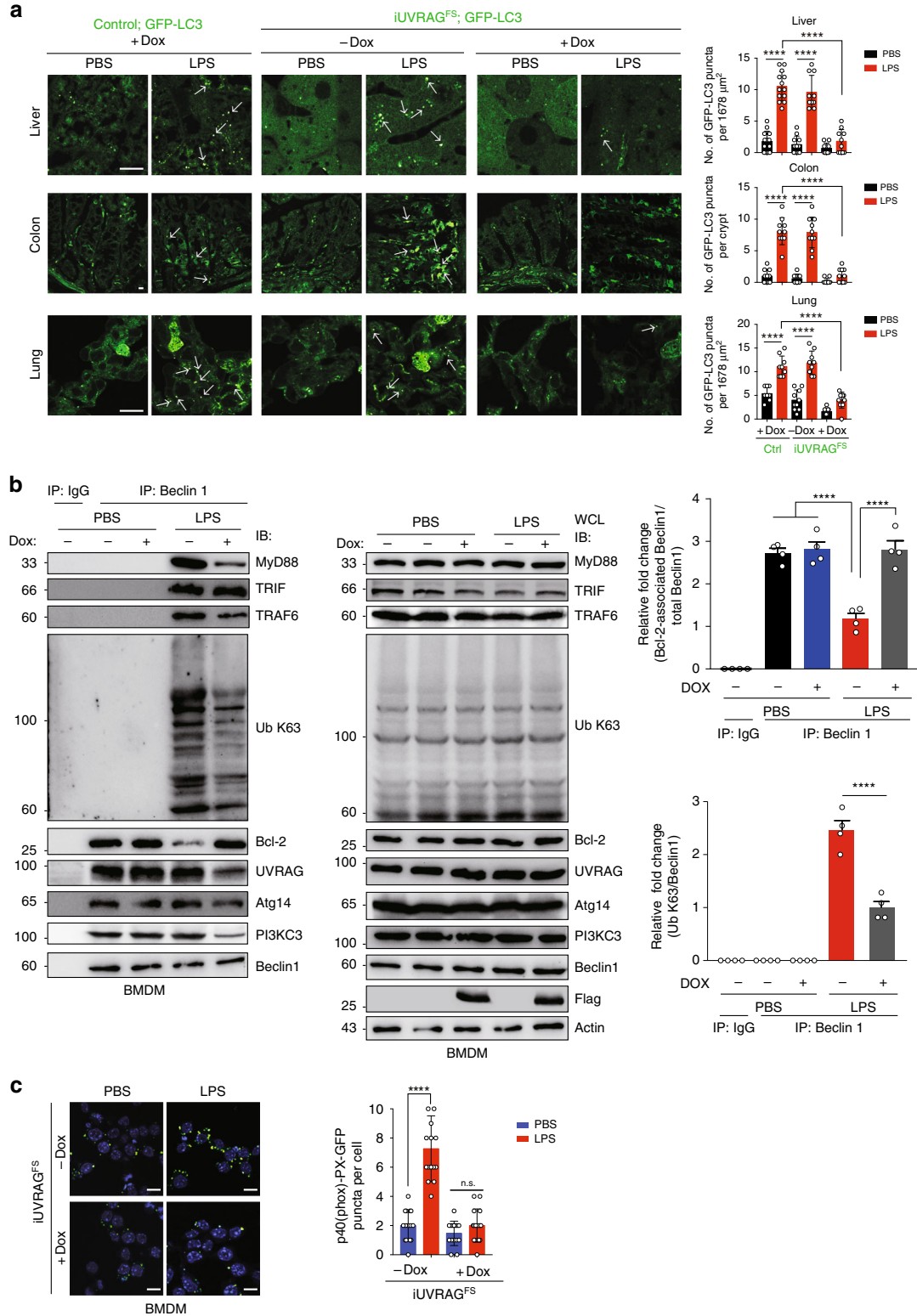

Despite NF-κB-dependent upregulation of NLRP3 by LPS + ATP, no differences in NLRP3 inflammasome protein expression were observed in control versus UVRAG[FS]-expressing cells (Fig. 3b), suggesting that increased NLRP3 inflammasome assembly by UVRAG[FS] is not due to altered protein expression. Administration of MCC950, a small-molecule inhibitor of NLRP3 inflammasome[39], decreased the levels of IL-1β in serum and improved the survival rate of mice treated with LPS (Fig. 3e,f),

suggesting that aberrant activation of NLRP3 inflammasome directly contributes to UVRAG[FS]-associated massive inflammation during sepsis. To examine whether the observed increase in NLRP3 inflammasome assembly/activation is due to specific effects of UVRAG[FS], or due to defective autophagy, we conducted the same experiment in macrophages from mice that were deficient for the autophagy gene Atg5 (Atg5[−/−]), an essential autophagy protein that associates with Atg12 and Atg16 to

**Fig. 2** UVRAG$^{FS}$ prevents TLR4 induction of autophagy in vivo. **a** Representative images (left panels) and quantification (right panels) of GFP-LC3 puncta in indicated tissues from Dox-treated/untreated iUVRAG$^{FS}$ mice and littermate control mice crossed with GFP-LC3 transgenic mice, challenged with PBS or LPS (20 mg per kg body weight; i.p.). Arrows denote GFP-LC3 puncta. Scale bars, 10 μm. $n = 10$ tissue sections pooled from three independent experiments. **b** UVRAG$^{FS}$ inhibits the Beclin1-autophagy interactome and K63-linked ubiquitination in LPS-stimulated bone marrow-derived macrophages (BMDMs). BMDMs derived from iUVRAG$^{FS}$ mice were incubated with PBS or LPS (100 ng/ml) for 4 h in the presence/absence of Dox and subjected to IP with anti-Beclin1 followed by WB of indicated proteins (left panel). Expression of indicated proteins in WCL were shown (middle panel). Densitometric quantification of the Bcl-2-associated Beclin1/total Beclin1 and the ubiquitinated (K63)-Beclin1/total Beclin1 ratios in BMDM are also shown (right panels). **c** UVRAG$^{FS}$ prevents LPS-induced PI3KC3 activation in BMDM. Cells cultured as in **b** were transfected with p40(phox)-PX-GFP and subjected to confocal microscopy. Representative images of p40(phox)-PX-GFP puncta in indicated BMDM are shown (left) and the number of p40(phox)-PX-GFP puncta per cell was quantified (right). Scale bars, 10 μm. $n = 10$–15 high-power fields (HPF) pooled from three independent experiments. Data in **b** are from one experiment that is representative of four independent experiments. For all quantifications, data (mean ± SD) were from the indicated number of independent experiments and analyzed with two-way ANOVA. n.s., not significant; ****$P < 0.0001$. Source data are provided as a Source Data file. See Supplementary Fig. 9 for uncropped data of **b**.

conjugate LC3 to autophagosome membrane[40]. As mice with constitutive Atg5 deletion die soon after birth[41], we used BMDM from inducible Atg5 knockout mice, whereby addition of tamoxifen abrogated Atg5 expression (Supplementary Fig. 3g). As seen with UVRAG$^{FS}$ expression, Atg5-deficient BMDM exhibited deficient autophagy activation, increased NLPR3:ASC: pro-caspase-1 complex formation, increased caspase-1 (but not caspase-11) activation, and increased IL-1β maturation compared with WT controls after LPS + ATP treatment (Supplementary Fig. 3g). The similarity of the phenotypes of cells from UVRAG$^{FS}$ mice and Atg5-deficient mice provides strong support for a role of suppressed autophagy, rather than other UVRAG$^{FS}$-regulated functions, in NLRP3 inflammasome activation and increased inflammatory response in iUVRAG$^{FS}$ mice.

Additionally, Dox-treated iUVRAG$^{FS}$ mice exhibited increased mitochondrial membrane potential (Δψm) loss (Supplementary Fig. 3h), cytosolic mitochondrial DNA (mtDNA) release (Supplementary Fig. 3i), and mtROS production (Fig. 3g) in LPS + ATP-treated BMDM. This increase was associated with impaired mitochondrial homeostasis, as indicated by increased aggregation of mitophagy marker protein p62[42] and Parkin, which selectively labels impaired mitochondria[43], on Tom20-labeled mitochondria (Fig. 3h and Supplementary Fig. 3j). Excess mtROS also resulted in increased genomic instability, as suggested by higher levels of γ-H2AX[44] in Dox-treated iUVRAG$^{FS}$ BMDM (Fig. 3b). Treatment of cells with the mitochondrial-specific antioxidant MitoQ[45], which eliminated not only mtROS but also prevented mtDNA release (Fig. 3g and Supplementary Fig. 3i), abrogated the increase of both caspase-1 activation and IL-1β production in Dox-treated iUVRAG$^{FS}$ BMDM in response to LPS + ATP (Fig. 3i). These data are in agreement with previous studies[5] demonstrating that autophagy, particularly mitophagy, prevents perpetuated inflammasome activation by eliminating inflammasome activators such as mtROS or mtDNA[38,46]. Notably, autophagy is also found to suppress cytosolic DNA-induced AIM2 inflammasome activation by promoting its lysosomal turnover[47]. However, UVRAG$^{FS}$-expressing BMDM responded normally to poly(dA:dT) treatment, when compared to control cells (Supplementary Fig. 3k), suggesting that UVRAG$^{FS}$-suppressed autophagic degradation might not play a dominant role in the modulation of AIM2 inflammasome. Nevertheless, these results indicate that the impaired LPS-induced autophagy activation is responsible for aberrant NLRP3-inflammasome activation and increased septic shock in iUVRAG$^{FS}$ mice.

**UVRAG$^{FS}$ exacerbates inflammatory responses in colitis.** Abnormal inflammasome activation is implicated in inflammatory bowel diseases (IBD)[48,49]. We next examined whether UVRAG$^{FS}$ suppression of autophagy exacerbates intestinal inflammation induced by oral administration of dextran sodium

sulfate (DSS), an epithelial irritant[50]. Dox-treated cohorts of WT and iUVRAG$^{FS}$ mice were fed 5% DSS (w/v) in their drinking water for 6 days, followed by 4 days of regular drinking water. Expression of UVRAG$^{FS}$ made mice more susceptible to DSS-induced colitis (Fig. 4a). In addition, UVRAG$^{FS}$-expressing mice exhibited increased disease severity, as demonstrated by elevated disease activity indices (DAI) and colonic shortening at necropsy on days 6 and 10 (Fig. 4b, c). Enhanced colitis corresponded with an increase in all indices of histopathological changes, including crypt damage and inflammatory infiltrates (Fig. 4d, e). This increased susceptibility of UVRAG$^{FS}$-expressing mice to colitis is not due to inherent differences at baseline, because no difference in colon length and crypt structure were observed between two genotypes (Supplementary Fig. 4a, b). The number of bacteria in the feces of both cohorts was also comparable (Supplementary Fig. 4c). To assess whether UVRAG$^{FS}$ suppression of autophagy contributes to DSS-induced intestinal inflammation, we analyzed autophagy activity in DSS-treated colons. As expected, DSS treatment led to elevated levels of autophagy, as measured by increased LC3-II conversion and p62 turnover, in control colon lysates, which was suppressed in Dox-treated iUVRAG$^{FS}$ mice (Fig. 4f, g). Deficient autophagy was associated with increased NLRP3 inflammasome assembly and culminated caspase-1 activation as observed in sepsis, enabling enhanced cleavage of pro-IL-1β and pro-IL-18 into their biologically active forms IL-1β and IL-18 in DSS-treated UVRAG$^{FS}$ mice colons (Fig. 4f–h). By contrast, formation of other colitis-relevant inflammasomes such as AIM2, NLRP6, and NLRC4 by DSS was comparable in colons between control and UVRAG$^{FS}$-expressing mice (Fig. 4f), highlighting a critical role of the NLRP3 inflammasome in UVRAG$^{FS}$-associated colitis. In accord, amounts of proinflammatory cytokines IL-1β and IL-18, but not IL-6 and TNF-α, produced by colonic tissue were increased in DSS-fed UVRAG$^{FS}$-expressing mice versus control mice (Fig. 4i). Supporting this, UVRAG$^{FS}$-expressing mice had elevated induction of mRNA expression for chemotactic chemokines including CXCL1, CXCL2, CXCL3, CCL2, CCL3, and CXCL10 that can be induced by IL-1β in colon (Supplementary Fig. 4d) and enhanced neutrophil infiltration (Fig. 4d). As observed in sepsis, UVRAG$^{FS}$ did not alter the response of mice to DSS with respect to NF-κB activation in colon (Supplementary Fig. 4e).

To further determine the contribution of hematopoietic UVRAG$^{FS}$ versus non-hematopoietic UVRAG$^{FS}$ in driving exacerbated colitis, we generated UVRAG$^{FS}$ bone marrow chimeras by reconstituting both control and UVRAG$^{FS}$-expressing recipients with bone marrow harvested from either control or UVRAG$^{FS}$-expressing donors, as previously described[51], and confirmed UVRAG$^{FS}$ expression in these chimeras (Supplementary Fig. 4f). We observed that both sets of chimeric mice were more hypersensitive to DSS-induced colitis compared with the control→control animals, as indicated by increased colon length reduction

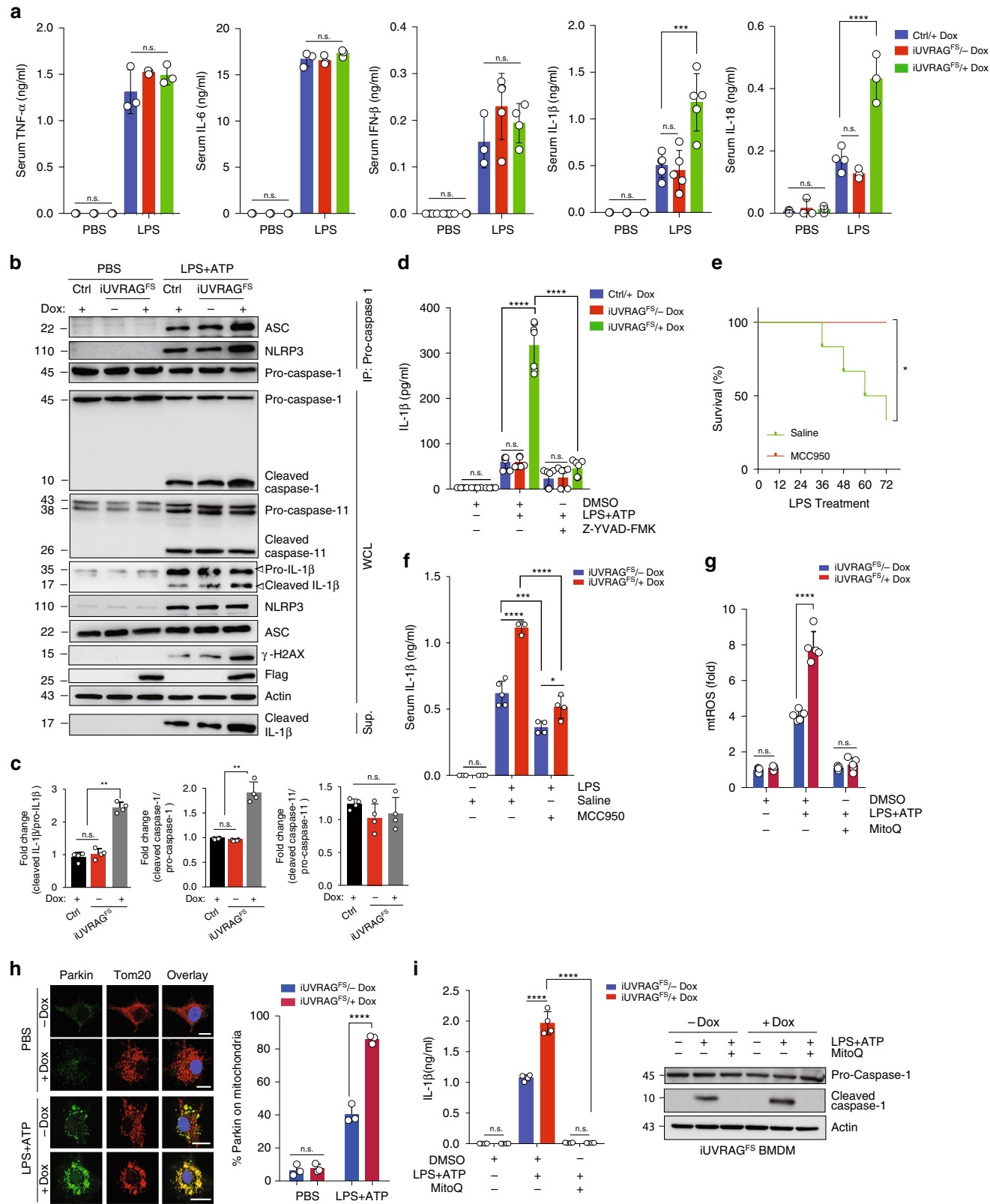

and histological indices (Supplementary Fig. 4g–i). Histopathological analysis of colons isolated from the chimeric mice revealed that all of the UVRAG^FS-expressing animals (both donor and recipient) exhibited increased crypt damage, extensive areas of ulceration, and enhanced inflammation compared with the *control→ control* mice (Supplementary Fig. 4h, i). Thus, UVRAG^FS functions through both hematopoietic and non-hematopoietic

compartments in inflammasome activation. Administration of the NLRP3 inhibitor MCC950[39] attenuated the clinical and histological signs of colitis in Dox-treated *iUVRAG^FS* mice, including weight loss, colon length reduction, histological indices, tissue injury, and inflammatory infiltrates (Supplementary Fig. 4j–n; Fig. 4j–l). Although inflammatory cytokines IL-6 and TNF-α were unaffected, colonic IL-1β was significantly lower after MCC950

**Fig. 3** UVRAG$^{FS}$ blockade of LPS-induced autophagy enhances inflammatory response. **a** ELISA of inflammatory cytokines in serum from *iUVRAG$^{FS}$* mice and control mice (*n* = 3–5 per group), assessed 3 h after LPS injection (20 mg per kg body weight; i.p.). **b** Co-IP of ASC and NLRP3 with pro-caspase-1 and WB analysis of indicated proteins in WCLs and supernatants (Sup.) of BMDM from *iUVRAG$^{FS}$* and control mice stimulated with PBS or LPS (100 ng/ml, 6 h) followed by 1 h of ATP (LPS + ATP). **c** Densitometric quantification of relative production of cleaved caspase-1, caspase-11, and IL-1β to their precursors in BMDM culture in **b**. **d** ELISA of IL-1β secretion by LPS-primed control and UVRAG$^{FS}$ BMDM incubated for 1 h with Z-YVAD-FMK or vehicle, followed by 1 h of ATP. *n* = 7. **e** Survival of *iUVRAG$^{FS}$* mice in response to MCC950 therapy in LPS-induced sepsis. Dox-treated *iUVRAG$^{FS}$* mice given LPS were treated with MCC950 (*n* = 4) or saline vehicle (*n* = 3–5) 1 h before LPS administration and then daily for 3 days, and their survival were plotted. **f** ELISA of serum IL-1β of Dox-treated/untreated *iUVRAG$^{FS}$* mice in response to MCC950 therapy in **e**. *n* = 3–5 mice. **g** Relative mtROS production of LPS-primed ATP-stimulated *iUVRAG$^{FS}$* BMDM in the presence/absence of MitoQ treatment (1 μM). *n* = 5. **h** Representative confocal microscopy (left) and quantification (right) of intracellular distribution of Parkin (green) relative to Tom20-labeled mitochondria (red) in LPS + ATP-treated *iUVRAG$^{FS}$* BMDM. *n* = 3. Scale bars, 10 μm. **i** ELISA of IL-1β secretion (left) and WB analysis for caspase-1 in lysates (right) of LPS-primed *iUVRAG$^{FS}$* BMDM incubated with MitoQ (1 μM, 1 h), followed by 1 h ATP. *n* = 4. Data in **b**, **i** are from one experiment that is representative of four independent experiments. For all quantifications, data (mean ± SD) were analyzed with two-way ANOVA. n.s., not significant; *$P < 0.05$; **$P < 0.01$; ***$P < 0.001$; ****$P < 0.0001$. Source data are provided as a Source Data file. See Supplementary Fig. 9 for uncropped data of **b**, **i**.

treatment (Fig. 4l). These results indicate that heightened NLRP3 activation with subsequently increased production of proinflammatory cytokines account for increased colitis in *iUVRAG$^{FS}$* mice.

**UVRAG$^{FS}$ predisposes mice to colitis-associated CRC.** We next examined whether autophagy restriction of inflammasome activation confers a protective role from colitis-associated neoplastic process in *iUVRAG$^{FS}$* mice, using an established azoxymethane (AOM)-DSS model of colorectal cancer (CRC)[52]. No difference in body weight was observed between Dox-treated *iUVRAG$^{FS}$* and control littermates (Supplementary Fig. 5b). However, we observed a more than two-fold increase in the incidence of tumors in Dox-treated *iUVRAG$^{FS}$* mice compared with controls, and the average tumor size and tumor load was elevated upon UVRAG$^{FS}$ expression (Fig. 5a–d). Immunohistochemistry analyses revealed increased cell proliferation (Ki67 staining) and decreased apoptosis, as indicated by reduced active caspase-3 staining, on day 60 post-treatment, in the colon of Dox-treated *iUVRAG$^{FS}$* versus controls (Fig. 5e). Furthermore, there were decreased levels of the epithelial cell markers, Keratin 20 (Krt20) and E-cadherin, but increased levels of the mesenchymal markers, N-cadherin and vimentin, in colons with UVRAG$^{FS}$ expression in the course of tumorigenesis, while the epithelial-to-mesenchymal transition (EMT)-related Wnt/β-catenin pathway remained unaffected (Fig. 5e and Supplementary Fig. 5c). The hyperproliferation and blocked differentiation by UVRAG$^{FS}$ is consistent with early onset of colitis-associated CRC in Dox-treated *iUVRAG$^{FS}$* mice.

In concordance with increased colonic tumorigenesis, Dox-treated *iUVRAG$^{FS}$* mice colon displayed severe crypt disruption and inflammatory infiltration after AOM-DSS treatment (Fig. 5f, g). There were increased caspase-1 activation and IL-1β production in the colon of Dox-treated *iUVRAG$^{FS}$* as compared to control mice, which correlated with suppressed autophagy, as measured by LC3 conversion and p62 turnover (Fig. 5e, h), supporting the concept that disturbed autophagy by UVRAG$^{FS}$ has a mechanistic role in unresolved inflammasome activation and inflammation-associated tumor susceptibility.

**Promotion of spontaneous tumorigenesis by UVRAG$^{FS}$.** Despite the oncogenic feature of UVRAG$^{FS}$ in vitro[16], it remained untested whether UVRAG$^{FS}$ was sufficient to initiate tumorigenesis in mammals. To this end, we aged cohorts of *iUVRAG$^{FS}$* mice and WT littermates that were fed Dox starting from 2 month of age and explored the impact of UVRAG$^{FS}$ on spontaneous tumorigenesis. Dox-treated *iUVRAG$^{FS}$* mice succumbed to the development of spontaneous tumors starting at 30 weeks (Fig. 6a). By 18 months of age, approximately 90% of

UVRAG$^{FS}$-expressing mice as compared with 32% of Dox-fed wild-type mice had malignancies (Supplementary Fig. 6a). The tumors in *iUVRAG$^{FS}$* mice on Dox comprised mainly of lymphomas and adenocarcinoma. We observed massive spleen and lymph node enlargement in most (70–80%) of UVRAG$^{FS}$-expressing mice autopsied at 6–18 months of age, which was histologically confirmed as lymphomas (Fig. 6b, c). By contrast, few lymphomas were observed in Dox-treated WT mice (Fig. 6c and Supplementary Fig. 6b). The lymphomas invaded a variety of tissues including the liver and lungs (Fig. 6b). The predominant type of lymphomas developed in UVRAG$^{FS}$-expressing mice was diffuse large B cell lymphoma (DLBCL), which showed strong immunoreactivity for the B cell marker, CD20, but were negative for the germinal center marker Bcl-6 (Fig. 6d). Some Dox-induced *iUVRAG$^{FS}$* mice also developed low grade B cell lymphoma, suggestive of extranodal marginal zone B cell lymphoma with crystal storing histiocytosis, positive for monotypic kappa (Fig. 6d). In addition to frank lymphomas, there was increased prevalence of splenic, nodal, and extranodal lymphoproliferative disease (LPD) associated with UVRAG$^{FS}$ expression (Fig. 6c). Moreover, Dox-treated *iUVRAG$^{FS}$* mice had lung lesions in earlier stages of neoplasia, including poorly differentiated adenocarcinomas with nuclear immunoreactivity for TTF-1, a tissue-specific transcription factor in lung epithelial cells[53] (Fig. 6d). All malignancies had detectable UVRAG$^{FS}$ expression in tumor cells (Fig. 6d). Thus, expression of this cancer-derived UVRAG$^{FS}$ results in increased susceptibility to spontaneous malignancies.

**UVRAG$^{FS}$ promotes proliferation and β-catenin activation.** Although colon tumors were not observed in our cohort of aged mice, histopathologic examination of colons from Dox-treated *iUVRAG$^{FS}$* mice revealed hyperplastic changes compared with those from controls (Fig. 7a). Crypt length was increased by ~1.3-fold along the entire length of the colon in Dox-treated *iUVRAG$^{FS}$* mice along with cytoplasmic expression of UVRAG$^{FS}$ at the colonic crypts (Fig. 7a, b). No significant differences were detected in the number of cleaved caspase-3-positive cells in the colon of control versus UVRAG$^{FS}$-expressing mice (Fig. 7c). However, there was increased Ki67 staining throughout the crypts in UVRAG$^{FS}$-expressing mice (Fig. 7c, d). The increased proliferation in colons from Dox-treated *iUVRAG$^{FS}$* mice was accompanied by a notable increase in genomic instability, as measured by levels of γ-H2AX[44] (Fig. 7c, d), providing in vivo support for a putative defect in DNA damage repair in cells with UVRAG$^{FS}$ expression[16]. We next asked whether the pro-proliferative effect of UVRAG$^{FS}$ in the colon of aged mice might also involve aberrant inflammasome activation, a risk factor for intestinal tumorigenesis[54]. No notable caspase-1 activation, IL-1β and IL-18 production were detected in both young

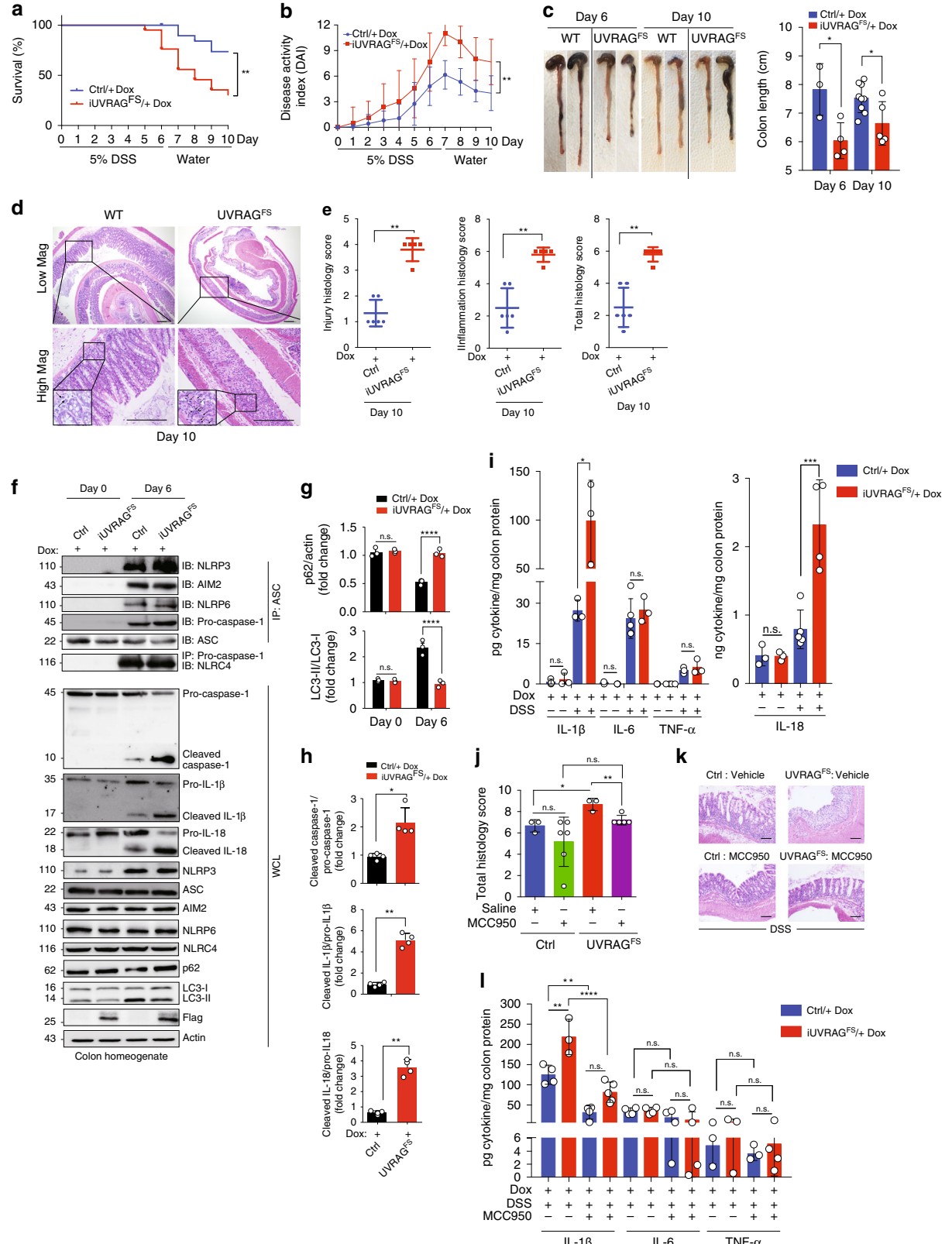

(2-month-old) and older (18-month-old) mice in either genotype (Supplementary Fig. 7a). Likewise, no discernable change was observed in the expression of inflammasome proteins (i.e. ASC, NLRP3, NLRP6, AIM2, and NLRC4) in these mice colons (Supplementary Fig. 7a). Similar results were obtained in splenocytes that gave rise to increased spontaneous malignancies

(Supplementary Fig. 7b). These data indicate that UVRAG[FS] affects cell proliferation and spontaneous tumorigenesis independently of inflammasome activation.

The increased cell proliferation in *iUVRAG[FS]* mice suggested that an UVRAG[FS]-dependent oncogenic signaling or mitogen might be involved in spontaneous tumorigenesis. Given the

**Fig. 4** UVRAG$^{FS}$ exacerbates intestinal colitis in vivo. **a** Kaplan–Meier curve showing survival of control and *iUVRAG$^{FS}$* mice (*n* = 10 per group) fed with 5% DSS in drinking water for 6 days. \*\**P* < 0.01 (Log-rank test). **b** Disease activity index (DAI) of colitis in control and *iUVRAG$^{FS}$* mice in **a**. The DAI value was calculated as described in Methods. **c** Colon lengths of DSS-treated mice with indicated genotypes were assessed at the time of necropsy (days 6 and 10) with representative macroscopic images (left). **d** H&E-stained sections of colon from DSS-treated mice with indicated genotypes on day 10 after DSS. Inset magnification highlights inflammation and crypt damage. Arrows indicate neutrophil infiltration. Scale bars, 300 μm. **e** Histopathology scores as indicated in the colon of mice with indicated genotypes after DSS-induced colitis. *n* = 5–6 per genotype. **f** Co-IP of inflammasome complex assembly as indicated in colon tissues of mice with indicated genotypes before/after DSS. WB of indicated protein expression in colon homogenates are shown (lower panel). **g**, **h** Densitometric quantification of the ratios of LC3-II/LC3-I and p62/actin (**g**) and the relative production of cleaved caspase-1, IL-1β, and IL-18 to their precursors (**h**) in the colons in **f**. *n* = 3 (**g**) and 4 (**h**) independent experiments. **i** ELISA of IL-1β, IL-6, and TNF-α (left panel) and IL-18 (right panel) in colons of mice with indicated genotypes after DSS treatment. *n* = 3–5 per genotype. **j**, **k** Histopathology scores (**j**) and H&E-stained sections (**k**) of colons from mice with indicated genotypes treated with MCC950 during DSS-induced colitis. *n* = 3–6 per genotype. Scale bars, 100 μm. **l** ELISA of cytokines in colons of mice with indicated genotypes in **j**. *n* = 3–5 per genotype. Data in **f** are from one experiment that is representative of three independent experiments. For all quantifications, data (mean ± SD) were analyzed with Student's *t*-test or two-way ANOVA. n.s., not significant; \**P* < 0.05; \*\**P* < 0.01; \*\*\**P* < 0.001; \*\*\*\**P* < 0.0001. Source data are provided as a Source Data file. See Supplementary Fig. 9 for uncropped data of **f**.

essential role of the Wnt/β-catenin pathway in intestinal homeostasis and tumorigenesis[55], we tested whether UVRAG$^{FS}$ regulates β-catenin expression. UVRAG$^{FS}$ expression resulted in an increase in overall β-catenin levels and in their nuclear accumulation in colon (Fig. 7e, f). Immunoblotting analysis of colonic β-catenin further confirmed increased expression of β-catenin and concomitant upregulation of β-catenin target genes, *c-Myc* and *Cyclin D1*, in 18-month-old UVRAG$^{FS}$-expressing mice but not in WT mice of the same age (Fig. 7g). Supporting this, Dox-treated *iUVRAG$^{FS}$* mice (18-month-old) had elevated induction of mRNA for β-catenin targets, while the mRNA abundance of β-catenin was not increased (Fig. 7h). Analogous results were obtained in splenocytes (Supplementary Fig. 7c). Notably, in the colon of young mice, no genotype-specific differences were observed in any of the β-catenin-related factors assessed (Fig. 7g), indicating that the enhanced β-catenin activation in older UVRAG$^{FS}$-expressing mice genuinely reflects age-related changes.

The increase in β-catenin protein without a concomitant increase in mRNA level in *iUVRAG$^{FS}$* mice suggested that β-catenin is subjected to post-translational regulation. Canonically, cytoplasmic β-catenin protein levels are controlled by the Axin1-APC-GSK3 destruction complex (DC), and mutations in DC components stabilizes and activates β-catenin, resulting in cancer, mostly notably in the colon[56]. Exon sequencing of *Axin1, APC*, and *GSK3* in colonic DNA from the 18-month-old Dox-treated *iUVRAG$^{FS}$* mice did not reveal any genetic changes. Both the young and older Dox-treated *iUVRAG$^{FS}$* mice and controls expressed comparable levels of DC components and the β-catenin DC complex assembly (Supplementary Fig. 7d). These results point to a role for UVRAG$^{FS}$ in β-catenin activation in older mice through a mechanism independently of the β-catenin DC properties, which induces transcription of its target oncogenes and increases cancer susceptibility.

**UVRAG$^{FS}$ activates β-catenin via autophagy suppression**. Autophagy was recently implicated in the regulation of Wnt/β-catenin pathway[57,58]. We postulated that UVRAG$^{FS}$ exerts its role in β-catenin activation by autophagy suppression. Although UVRAG$^{FS}$ did not readily affect basal autophagy in young mice, it exacerbated age-related decline in autophagic function in 18-month-old mice in conjunction with increased β-catenin activity in vivo (Fig. 7g and Supplementary Fig. 7c). Inhibition of autophagy by 3-methyladenine (3-MA), chloroquine (CQ), or by depletion of Beclin1 and UVRAG, increased β-catenin protein levels in immortalized fetal human colon epithelial cells (iFHC) and in SW480 CRC cells (Fig. 7i, j and Supplementary Fig. 7e, f), indicating a steady-state turnover of β-catenin through the autophagy-lysosome pathway. We therefore tested whether UVRAG$^{FS}$ expression is sufficient to stabilize β-catenin protein in

colon epithelial cells. Concomitant with the dose-dependent decrease of autophagy in UVRAG$^{FS}$-expressing SW480 cells, there was an increase in β-catenin protein levels and the transcriptional output of the Wnt pathway, accordingly, while β-catenin mRNA levels remained unaffected (Fig. 7k, l). UVRAG$^{FS}$ expression also promoted the G1/S transition and cell cycle progression as expected (Fig. 7m). Indeed, elevated β-catenin proteins correlated with a marked decrease in β-catenin association with the autophagy marker proteins LC3-II (Fig. 7k). Conversely, overexpression of WT UVRAG or treating cells with Torin 1, which reversed the decreased autophagy induced by UVRAG$^{FS}$, promoted β-catenin association with LC3-II and reduced β-catenin levels to those observed in control cells (Supplementary Fig. 7g). A similar reduction in β-catenin turnover and consequent upregulated β-catenin target-gene expression were also observed in SW480 cells with UVRAG knockdown (Supplementary Fig. 7h, i), suggesting that UVRAG suppression, rather than other potential effects of UVRAG$^{FS}$, contributes to β-catenin regulation. Consistently, activation of β-catenin by UVRAG suppression was associated with an increase in clonogenic survival of SW480 cells (Supplementary Fig. 7j), which was reversed by iCRT14, an inhibitor of β-catenin-TCF binding[59]. Hence, the inhibited clonogenic growth by iCRT14 supported the regulatory role of β-catenin on tumorigenesis associated with UVRAG suppression and suggest that decreased autophagy is a major contributor to aberrant β-catenin activation in non-challenged conditions. We confirmed that these effects of autophagy suppression in vitro on β-catenin regulation also occurred in vivo. In 18-month-old *iUVRAG$^{FS}$* mice, induced UVRAG$^{FS}$ expression decreased autophagy and resulted in increased levels of β-catenin (Fig. 7n and Supplementary Fig. 7k). The reduced association of β-catenin with autophagosomes in these mice was accompanied by increased interaction with TCF4, leading to upregulation of the Wnt/β-catenin signaling (Fig. 7n and Supplementary Fig. 7k). These results indicate that UVRAG$^{FS}$ stabilizes β-catenin by inhibiting it autophagic degradation, a mechanism that may contribute to spontaneous tumorigenesis in UVRAG$^{FS}$-expressing mice.

**Centrosome amplification associated with UVRAG$^{FS}$**. Because the role of UVRAG$^{FS}$ in cancer is also linked to its ability to promote centrosome amplification (CA) and chromosomal instability in vitro[16], we further investigated the effect of UVRAG$^{FS}$ on centrosome homeostasis in vivo. There was a significant increase in the incidence and degree of supernumerary centrosomes in the tissues of 18-month-old Dox-treated *iUVRAG$^{FS}$* mice as compared to the age-matched control mice (Fig. 8a–c). This phenotype was most pronounced in the colon and spleen of *iUVRAG$^{FS}$* versus control mice (Fig. 8a, b), where high rates of proliferation was observed (Figs. 6 and 7c). In young

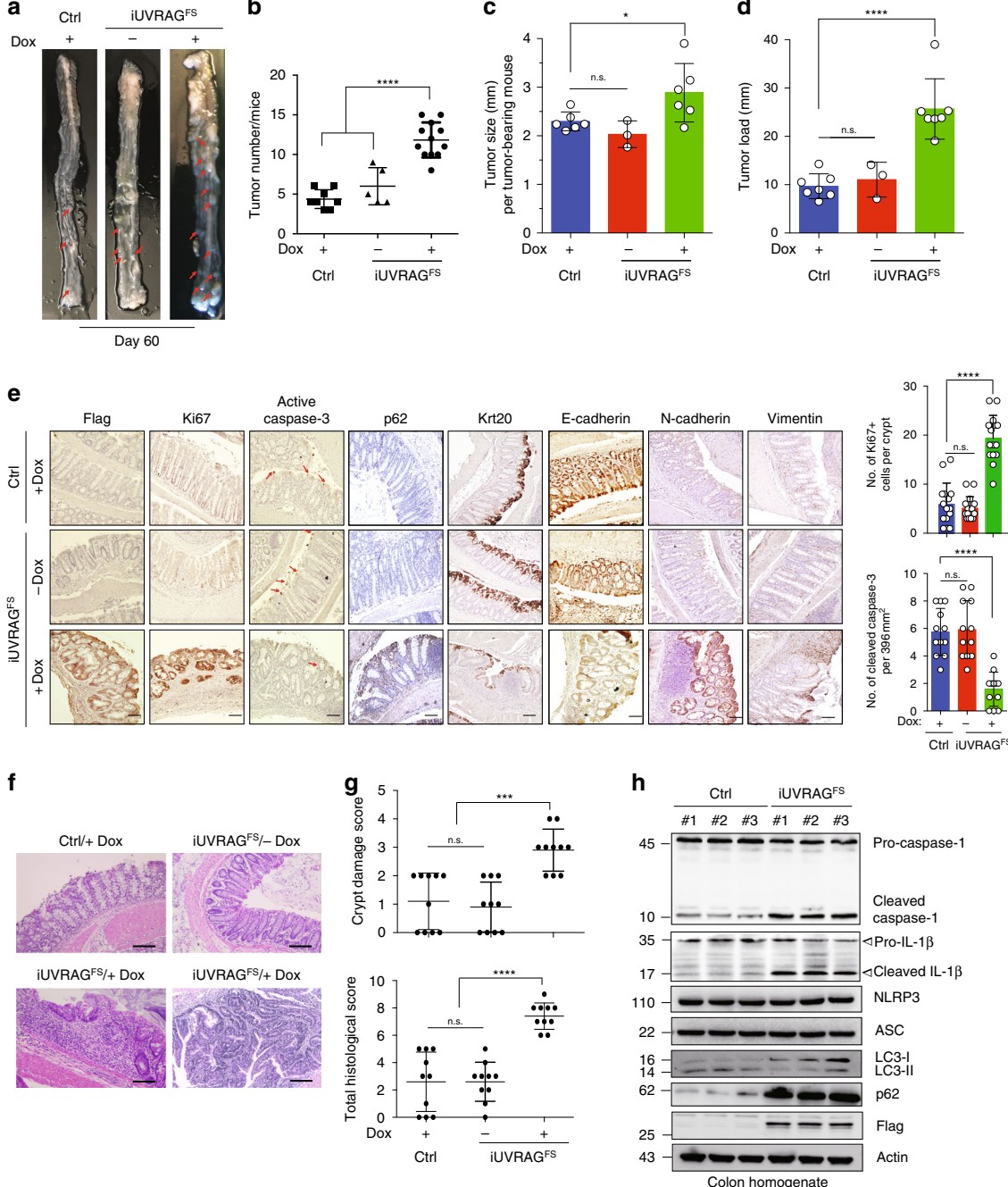

**Fig. 5** UVRAG[FS] predisposes mice to colitis-associated colon cancer. **a** Representative macroscopic images of the colon from AOM-DSS-treated control (Ctrl) and *iUVRAG[FS]* mice in the presence/absence of Dox at the time of necropsy (day 60). Red arrows highlight colonic tumors. **b–d** The number (**b**), size (**c**), and overall load (**d**) of colonic tumors in AOM-DSS-treated mice in **a**. Tumor load is evaluated by totaling the diameters of all tumors in AOM-DSS-treated mice in **a**. $n = 5–12$ mice per genotype. **e** IHC staining of Ki67, cleaved caspase 3, p62, keratin 20, E-cadherin, N-cadherin, and vimentin in the colons from AOM-DSS-treated mice of indicated genotypes. Data are from one animal that is representative of 5–12 animals in each group. The levels of Ki67 staining (top right) and tumor apoptosis (bottom right) in the indicated colon were quantified. Arrows (red) denote cells undergoing apoptosis. Scale bars, 100 μm. **f** H&E-stained sections of the colon from AOM-DSS-treated mice in **a**. Data are from one animal that is representative of 5–12 animals in each group. Scale bars, 100 μm. **g** Histological scores for crypt damage (top) and total histological score (bottom) of colons from mice in **a** were quantified at day 60 as described in the Methods. $n = 10$ mice per group. **h** WB analysis of caspase-1 cleavage, IL-1β production, LC3-I/II, and p62 levels in colon tissues of Dox-treated control and *iUVRAG[FS]* mice after AOM-DSS treatment. Data in **h** are from one experiment that is representative of three independent experiments. For all quantifications, data (mean ± SD) were from the indicated number of independent experiments and analyzed with one-way ANOVA. n.s., not significant; *$P < 0.05$; ***$P < 0.001$; ****$P < 0.0001$. Source data are provided as a Source Data file. See Supplementary Fig. 9 for uncropped data of **h**.

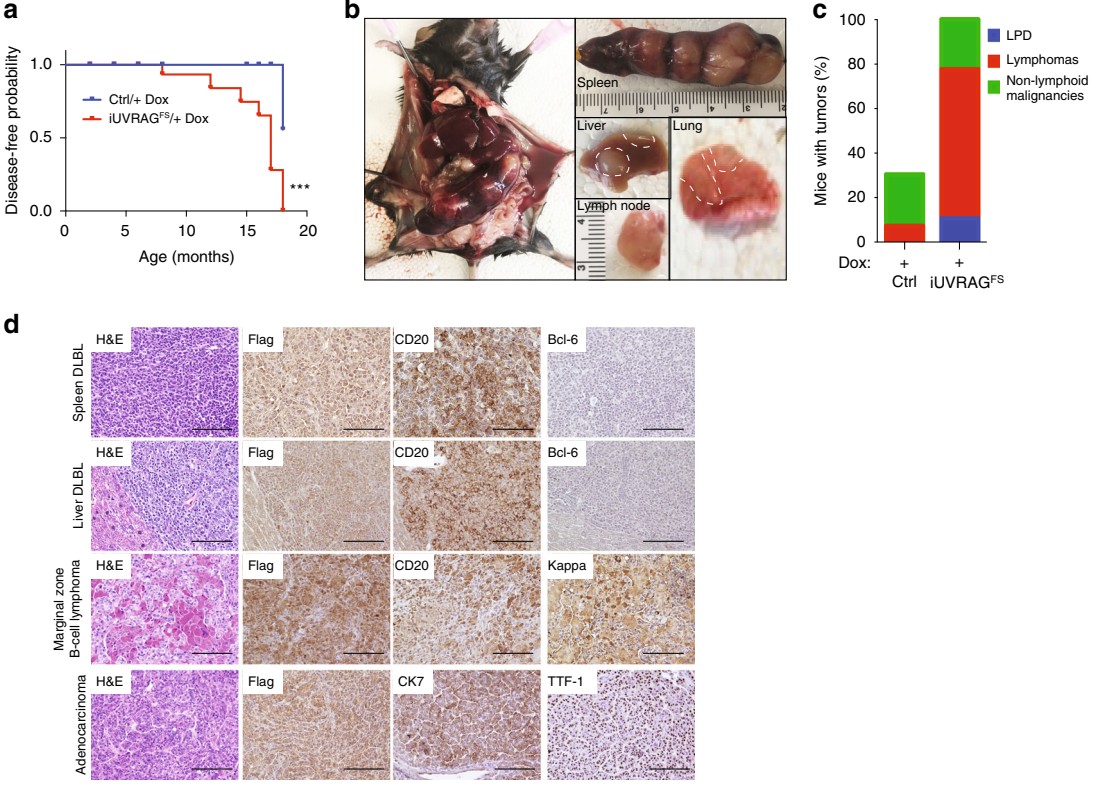

**Fig. 6** UVRAG$^{FS}$ promotes spontaneous tumorigenesis in mice. **a** Kaplan–Meier plot of time to development of any malignancy in Dox-treated *iUVRAG$^{FS}$* mice (*n* = 25) versus Dox-treated control mice (*n* = 25). Malignancy is determined by complete histologic survey of all major internal organs. ***P < 0.001 (Log-rank test). **b** Representative images of DLBCL in different organs that developed in Dox-treated *iUVRAG$^{FS}$* mice. **c** Percentage of control (*n* = 16) and UVRAG$^{FS}$-expressing (*n* = 12) mice with spontaneous tumors, including lymphoproliferative disease (LPD), lymphomas, and non-lymphoid malignancies. **d** H&E and IHC analysis of the most frequently observed neoplastic lesions from Dox-treated *iUVRAG$^{FS}$* mice using the antibodies as indicated. Scale bars, 100 μm.

(2-month-old) mice, despite a modest elevation in the level of CA upon UVRAG$^{FS}$ expression, no genotype-specific differences were observed with statistical significance in any of the tissues assessed (Supplementary Fig. 8a, b). This indicates that the exacerbated centrosome pathology in older UVRAG$^{FS}$-expressing mice also reflects an increase in age-related changes in these organs. Consistent with the consensus that CA causes erroneous chromosomal segregation[60], we detected a greater than 3-fold increase in aneuploidy in UVRAG$^{FS}$-expressing splenocytes from both 18-month-old mice and 2-month-old mice to lesser extent (Fig. 8d). In parallel, genome-wide copy number variation (CNV) analysis[61] of tumors developed in Dox-treated *iUVRAG$^{FS}$* mice revealed marked variability and heterogeneity with more chromosomal amplifications and deletions than the control (Fig. 8e). Taken together, these data indicate that CA induced by UVRAG$^{FS}$ may play a role in UVRAG$^{FS}$-associated chromosomal aneuploidies that potentially favor tumor formation in UVRAG$^{FS}$-expressing mice.

## Discussion

Herein, we demonstrate that a transgenic frameshift mutation in a key autophagy regulatory gene, *Uvrag*, which disrupts the UVRAG autophagy-regulatory complex in mice, results in impaired starvation- or endotoxin-induced autophagy activation, increased inflammatory response and associated pathologies, as well as age-related spontaneous malignancies. Thus, UVRAG has previously undescribed essential roles in the in vivo regulation of stimulus-induced autophagy as well as inflammatory disorders. Furthermore, our discovery indicates that time-dependent

autophagy suppression by UVRAG$^{FS}$ and ensuing β-catenin stabilization/activation may serve as one tumor-promoting mechanism underpinning age-related cancer susceptibility.

Due to the embryonic lethality of *Uvrag* knockout mice, the exact in vivo role of mammalian UVRAG in autophagy regulation remained largely unaddressed. Using the GFP-LC3 reporter to track autophagy and an inducible *Uvrag* mutant mouse model, we demonstrated that the cancer-derived dominant negative mutant of UVRAG blocks starvation-induced autophagy because Dox-induced *iUVRAG$^{FS}$* mice accumulated fewer autophagosomes upon starvation and displayed reduced rates of autophagic flux. We also found that lipophagy was impaired in UVRAG$^{FS}$ mice. Notably, our observation does not exclude a role for UVRAG$^{FS}$ and by extension autophagy inhibition in dysregulating other processes of lipid consumption (e.g. lipolysis or lipid oxidation), which merits further investigation. Likewise, we cannot definitely exclude other anti-autophagy effects of the UVRAG$^{FS}$ mutant besides UVRAG inhibition. Our data with *iUVRAG$^{FS}$* mice also reveal that the effect of UVRAG$^{FS}$ on basal autophagy was not as strong as that induced by starvation, implying that the UVRAG-associated PI3KC3 activity is either not essential, or alternatively, other PI3KC3 complexes, for instance one that assembled with Atg14, is sufficient for the basal levels of autophagy.

It is worth noting that Dox-induced *iUVRAG$^{FS}$* mice are also impaired in endotoxin-induced autophagy and are susceptible to death after LPS injection. This increased susceptibility was associated with NLRP3 inflammasome hyperactivation and increased inflammatory cytokine production. Primary BMDM from Dox-induced *iUVRAG$^{FS}$* mice, primed with LPS and treated with ATP, showed increased mtROS, NLRP3 inflammasome assembly and

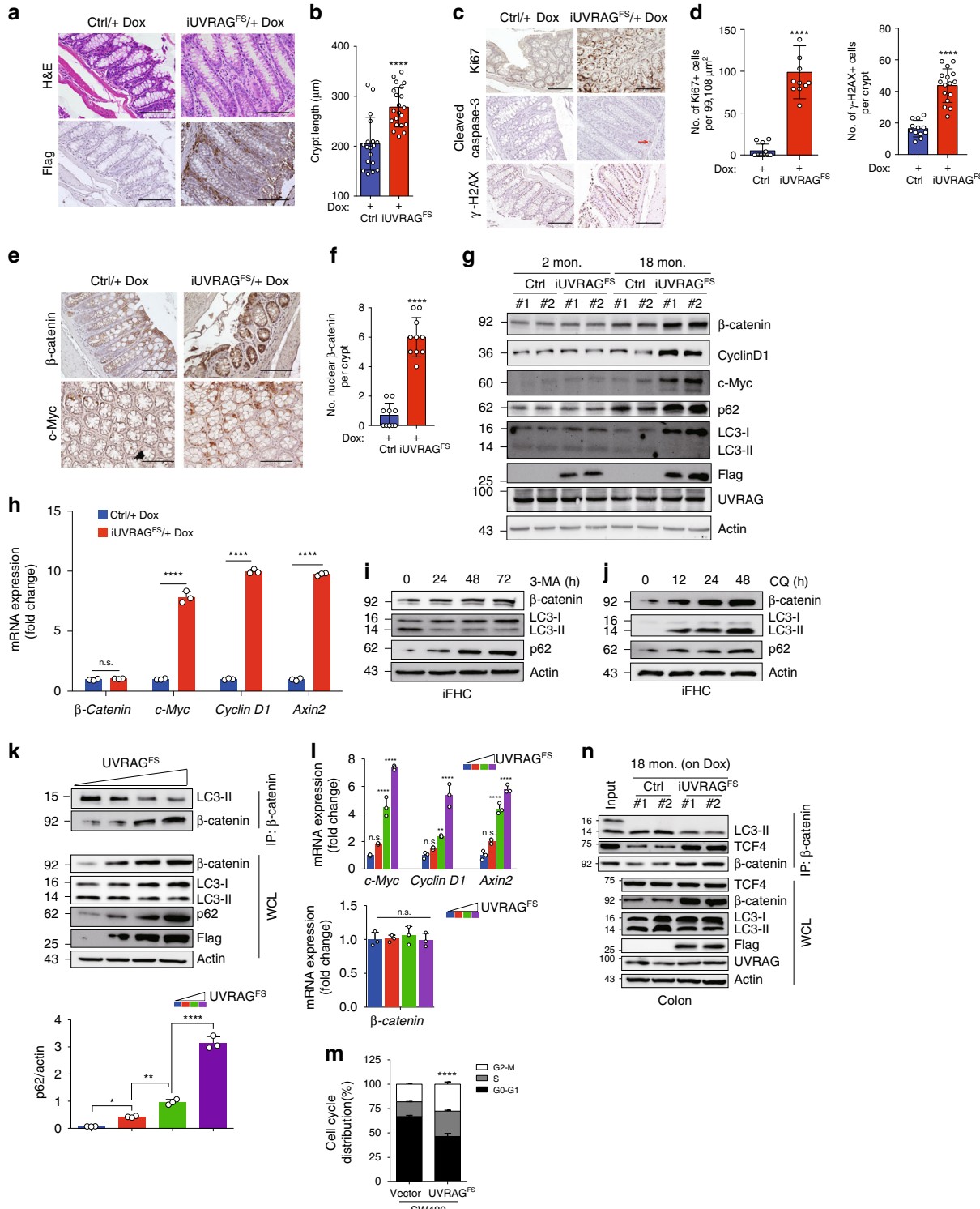

pro-caspase-1 cleavage, and hypersecretion of IL-1β. These phenotypes in UVRAG[FS] mice are consistent with those observed in previous studies of mice with deficiency of Atg16, LC3, or Beclin1[4,28]. It is difficult to conclude whether UVRAG functions directly through mitophagy or indirectly through other cellular factors in inflammatory response. However, the diminished recruitment of Beclin1 to the TLR4-receptor complex and the decreased UVRAG-Beclin1-Vps34 assembly/activity in Dox-induced iUVRAG[FS] mice suggest that the autophagy function of UVRAG is intrinsically required for mitochondrial homeostasis

and adequate regulation of inflammasome activation during sepsis. We also provide data showing that uncontrolled inflammation in iUVRAG[FS] mice was associated with increased tumor growth and de-differentiation in colitis-associated colon cancer.

In addition to tumor-promoting inflammation, increased cancer incidence is probably the most well-known phenotype of cells that are deficient in autophagy-related gene function[62,63]. However, the precise mechanisms underlying autophagy-related tumor prevention remains elusive. Although knockout of UVRAG is embryonic lethal in mice, the inactivation of UVRAG by the

**Fig. 7** UVRAG[FS] activates β-catenin by promoting age-related autophagy suppression. **a** H&E and IHC of colon from Dox-treated control and iUVRAG[FS] mice (12-month-old). Data are from one animal that is representative of 5–12 animals in each group. **b** Crypt length of the colon in **a**. **c**, **d** Representative images (**c**) of Ki67, cleaved caspase 3, and γ-H2AX of colons in 12-month-old control and iUVRAG[FS] mice on Dox and quantitation (**d**) of Ki67- and γ-H2AX-positive cells per crypt of indicated genotype. Arrow indicates apoptosis. **e**, **f** IHC of β-catenin and c-Myc (**e**) and quantitation of nuclear β-catenin per crypt (**f**) from control and iUVRAG[FS] mice (12-month-old). **g** WB of indicated protein expression in colons of the indicated genotype. **h** Quantitative RT-PCR of β-catenin and its target gene expression in colons from 18-month-old mice. n = 3. **i**, **j** WB of β-catenin and autophagy marker proteins in iFHC cells treated with 3-MA (1 mM) (**i**) or with chloroquine (CQ, 20 μM). **k** UVRAG[FS] inhibits β-catenin interaction with autophagy proteins. SW480 cells were transfected with increasing amounts of Flag-UVRAG[FS]. WCL were used for IP with anti-β-catenin, followed by IB with the indicated antibodies. The densitometric quantification of the p62/actin ratio is shown (bottom). **l** Quantitative RT-PCR of indicated gene expression in cells in **k**. n = 3. **m** Cell cycle analyses of SW480 cells stably expressing vector or UVRAG[FS]. n = 3. **n** Co-IP of β-catenin with LC3 and TCF4 in spleens from 18-month-old mice. Data in **b**–**f** are from 10 to 20 HPF pooled from three independent experiments. Data in **g**, **n** are from two randomly chosen samples per group with similar results observed in all ten samples per genotype. For all quantifications, data (mean ± SD) were analyzed with Student's t-test (**b**, **d**, **f**), two-way ANOVA (**h**, **m**) or one-way ANOVA (**k**, **l**, **m**). n.s., not significant; *P < 0.05; **P < 0.01; ***P < 0.001; ****P < 0.0001. Scale bars, 100 μm. Source data are provided as a Source Data file. See Supplementary Fig. 9 for uncropped data of **g**, **i**, **j**, **k**, **n** and Supplementary Fig. 10 for raw data of **m**.

expression of UVRAG[FS] causes the premature onset of age-related neoplasia, particularly spontaneous lymphoma, in iUVRAG[FS] mice, independently of its role in inflammasome regulation. Previous work has shown that human cancers, especially those with microsatellite instability, show a global expression of UVRAG[FS] compared with normal tissues[16]. Our work confirmed that this mutant promotes tumorigenesis in vivo. Intriguingly, UVRAG[FS] enhanced age-related decline of autophagy, which stabilized and thus increased the abundance of the β-catenin oncoprotein. These findings correlates age-related β-catenin accumulation with increased tumorigenesis in iUVRAG[FS] mice. Notably, overexpression/activation of β-catenin has been clinically correlated to the pathogenesis of DLBCL, a major tumor type developed in aged UVRAG[FS] mice, representing a new target for DLBCL lymphomagenesis[64,65]. More studies are needed to determine whether this age-dependent regulation of β-catenin by UVRAG[FS] can be generalized to other autophagy-related tumor models. Furthermore, considering the pleiotropic effects of UVRAG in cellular homeostasis, we cannot rule out other pro-tumorigenic effects caused by UVRAG inhibition other than the autophagy-Wnt/β-catenin pathway that may influence the tumorigenic phenotype in iUVRAG[FS] mice. For instance, we have provided genetic proof of the previously described roles for UVRAG[FS] in the regulation of CA and ensuant chromosomal instability[16], which may improve cancer cell fitness during tumor progression. Nevertheless, the emerging facet of oncogene-autophagy interactions highlight a functional role for autophagy, more likely selective autophagy, in age-related diseases including cancer.

## Methods

**Mouse models**. For generation of iUVRAG[FS] transgenic mice, a Flag-tagged human UVRAG[FS] mutant[16] was PCR amplified and subcloned into Mlu I and Not I restriction sites of pTRE-Tight (631059, Clontech), which contained a Tet-responsive $P_{tight}$ promoter and a SV40 poly(A). The IRES-luciferase (Luc) cDNA fragment was cloned into Not I site of pTRE-Tight. The resulting plasmid, pTRE[tight]-Flag-UVRAG[FS]-Luc, was digested with Xho I to release the transgenic cassette. The gel-purified cassette was injected into the pronucleus of fertilized 1-cell stage embryos (B6D2F1 background) with standard procedure. Injected embryos were cultured in M16 medium (M6111, Cytospreen) at 37 °C under 5% $CO_2$ overnight. All the two-cell stage embryos were then transferred into oviducts of the pseudopregnant CD-1 female mice at 0.5 dpc by Norris Comprehensive Cancer Center Transgenic Mice Core Facility (USC). Integration of the construct was confirmed by PCR (Supplementary Table 1). Two independent founder lines were identified and back-crossed for more than 20 generations to C57BL/6 mice (Jackson Laboratories). Flag-UVRAG[FS]-Luc transgenic mice were crossed with Rosa26-rtTA*M2 mice (Jackson Laboratories) in a pure C57BL/6 background to generate the double-transgenic mice (Rosa26-rtTA*M2;Flag-UVRAG[FS]-Luc), denoted as iUVRAG[FS]. Animals were maintained on the C57BL/6 background. To turn on the expression of UVRAG[FS], iUVRAG[FS] mice were administered a doxycycline (Dox) diet (TD.01306, Envigo) beginning at 22 days of age. iUVRAG[FS] mice and its wild-type littermate control mice were further crossed with GFP-LC3 transgenic mice[21] on the C57BL/6 background and tissues of offspring were used

for autophagy analyses in vivo. Both male and female control and iUVRAG[FS] mice were used in the studies.

Atg5[flox/flox] mice[21,65] were a gift from Dr. Noboru Mizushima (Tokyo Medical and Dental University, Tokyo, Japan). Atg5[flox/flox] mice were bred to Rosa26CreER[T2] mice (004847; Jackson laboratories) to generate Atg5[−/−] mice upon administration of tamoxifen.

For in vivo bioluminescence imaging, in vivo bioluminescence imaging was performed using the IVIS Lumina LT Series III in vivo imaging system (CLS136334, PerkinElmer)[66]. Images were captured and analyzed with Living Image® software. Signal intensity y was measured over the region of interest and quantified as flux (photons per s per $cm^2$ per sr).

For generation of bone marrow chimeric mice, bone marrow in recipient mice was ablated by lethal irradiation with 550 cGy (twice, 24 h apart)[67] before transplantation. Bone marrow were flushed from the femurs and tibias from wild-type control and iUVRAG[FS] donor mice on Dox and washed twice with warm PBS. In all, $1 × 10^7$ BM cells per mouse were infused intravenously into the tail veins of recipient mice. Mice were housed in microsolator cages for 6 weeks for full reconstitution and recovery before induction of DSS-colitis. Mice were provided with water containing gentamycin sulfate (0.2 mg/ml) for the first two weeks after transplantation. To assess BM reconstitution, peripheral blood was collected from chimeric mice and RNA was isolated using the Mouse RiboPure™ Blood RNA Isolation Kit according to the manufacturer's protocol (AM1951, Invitrogen) for RT-PCR with primers for mouse luciferase (Supplementary Table 2).

For LPS sepsis model, 8–10-week-old and sex-matched iUVRAG[FS] mice were injected intraperitoneally with 20 mg kg[−1] body weight of lipopolysaccharides (LPS) from E. Coli (L8274, Sigma). Tissues and blood were collected 3 h post injection. Survival after LPS challenge was assessed every 12 h for 3 days. All survived mice were euthanized at the end of the third day. To inhibit the NLRP3 inflammasome activation, control and iUVRAG[FS] mice on Dox were treated i.p. with 20 mg kg[−1] MCC950[39] or saline vehicle 1 h before LPS administration and then daily for 3 days. Mice were euthanized 24 h after the last treatment on day 3.

For induction of DSS-induced colitis and treatment studies, experimental colitis was induced by adding DSS (5% wt/vol) to the drinking water for 6 days, followed by a 4-day recovery period with water. Mice were weighed daily and monitored for clinical signs of colitis (e.g., weight loss, stool consistency, and rectal bleeding). To inhibit the NLRP3 inflammasome activation in vivo, control and iUVRAG[FS] mice on Dox were treated i.p. with 20 mg kg[−1] MCC950[39] or saline vehicle daily on days 0–5 of DSS administration. Mice were euthanized 24 h after the last treatment on day 6. A colitis disease activity index (DAI) was calculated for each mouse daily based on the following criteria[68]: weigh loss from baseline (0, no weight loss; 1, 1–3% weight loss; 2, 3–6% weight loss; 3, 6–9% weight loss; 4, >9% weight loss); stool consistency (0, normal; 2, loose stool; 4, diarrhea), and fecal blood (0, none; 2, blood visible in stool; 4, gross bleeding). Gross bleeding was defined as fresh perianal blood with obvious hematochezia. Upon necropsy, colon length was measured.

For induction of colitis-associated tumorigenesis, colitis-associated colon tumorigenesis was induced according to the literature[52]. Briefly, Dox-treated/untreated control and iUVRAG[FS] mice were administered AOM (10 mg kg[−1] body weight) intraperitoneally on day 0, followed by three cycles of DSS (2.5%, wt/vol) in the drinking water for five days with a 14-day water interval between each DSS cycle. The animals were weighed daily and sacrificed on day 60.

All mice were maintained in a pathogen-free facility with ad libitum access to food and water. All animal experiments were approved by the Institutional Animal Care and Use Committee (IACUC) of the University of Southern California and performed in accordance with IACUC guidelines for animal care and use.

**Cell culture and transfection**. 293T cells were cultured in Dulbecco's modified Eagle's medium (DMEM) (5796, Sigma). Primary fetal human colon epithelial cells (FHC) (CRL-1831, ATCC) were cultured in DMEM:F-12 medium, supplemented with 25 mM HEPES, 10 ng ml[−1] cholera toxin, 5 μg ml[−1] insulin, 5 μg ml[−1]

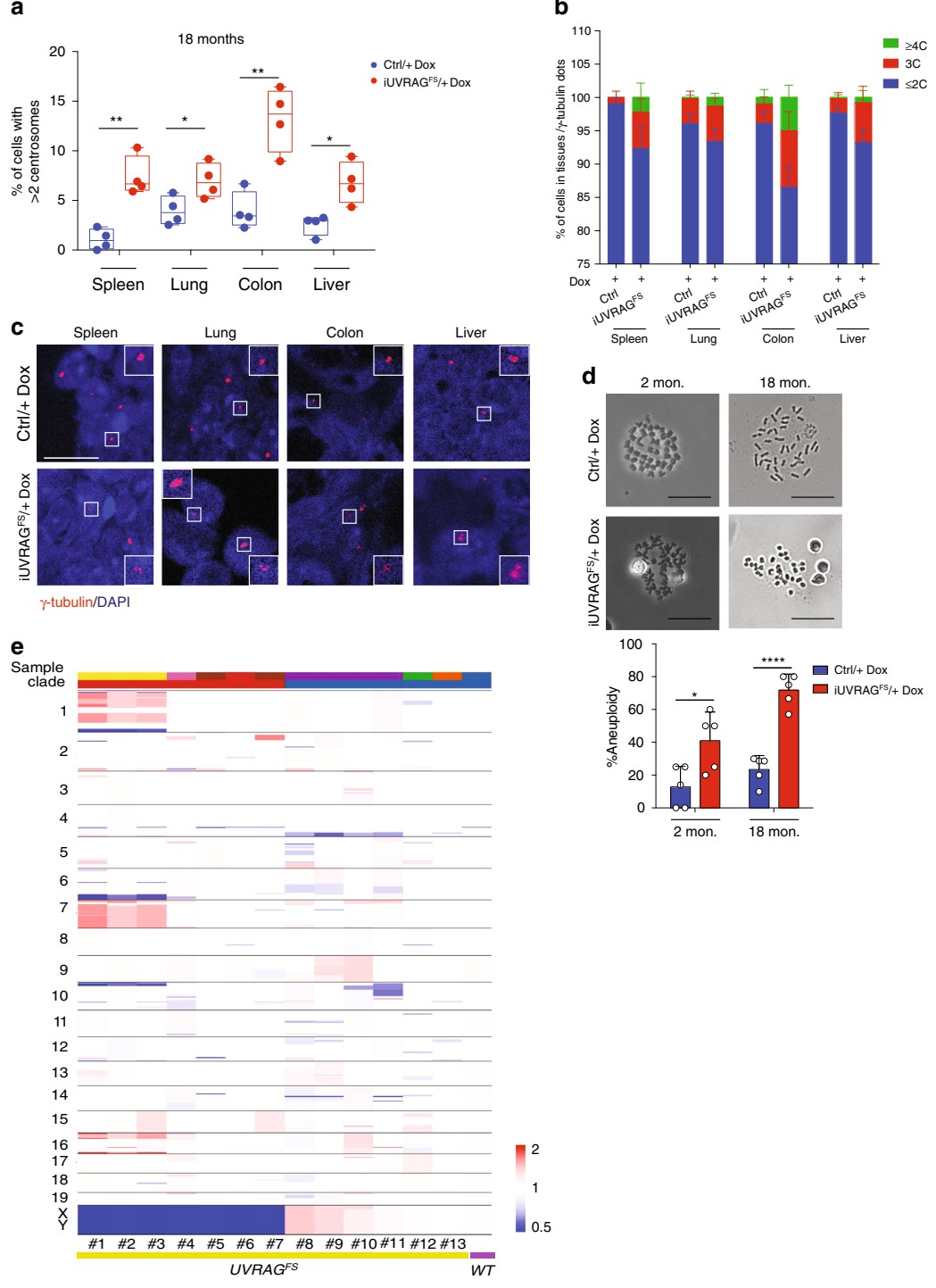

**Fig. 8 UVRAG$^{FS}$ promotes centrosome amplification in tissues. a** Quantification of the level of centrosome amplification in indicated tissues from Dox-treated 18-month-old control ($n = 4$) and *iUVRAG$^{FS}$* mice ($n = 4$). **b** Quantification of centrosome numbers in indicated tissues from 18-month-old Dox-treated control ($n = 4$) and *iUVRAG$^{FS}$* mice ($n = 4$). C, centrosome. **c** Representative confocal images of centrosomes in tissues from Dox-treated *iUVRAG$^{FS}$* mice or control animals. Centrosomes were immunostained for γ-Tubulin (red). Nuclei were stained with DAPI (blue). Data are from one animal that is representative of four animals in each group. Scale bar, 10 μm. **d** Representative images of metaphase spread from splenocytes of control ($n = 5$) and *iUVRAG$^{FS}$* mice ($n = 5$) treated with Dox for 2 or 18 months (left). The percentage of cells with abnormal karyotype (aneuploidy) was quantified (bottom). In all, 50–100 cells were evaluated per mice per experiment. Scale bars, 25 μm. **e** Heatmap of copy number alteration (CNA) profiles shows heterogeneous amplifications (red) and deletions (blue) among different tumors ($n = 13$) from Dox-treated *iUVRAG$^{FS}$* mice. CNA profiles are created using the mouse genome (mm9) as reference. Copy number is displayed as the ratio to the median. For all quantification, data represents the mean ± SD derived from the indicated number of independent experiments. Source data are provided as a Source Data file. *$P < 0.05$; **$P < 0.01$; ****$P < 0.0001$ (Student's *t* test).

transferrin, 100 ng ml$^{-1}$ hydrocortisone, and 20 ng ml$^{-1}$ human recombinant EGF. SW480 was cultured in Leibovitz's L-15 in the absence of $CO_2$. All media above were supplemented with 10% fetal bovine serum (FBS) (Seradigm), 2 mM L-glutamine, and 1% penicillin-streptomycin (Gibco-BRL). Murine fibroblast L-929 cells were purchased from ATCC (CRL 6364) and cultured in DMEM containing 10% Tet system approved FBS (631106, Clontech), 2 mM L-glutamine, and 1% penicillin-streptomycin (Gibco-BRL). Transfections were performed using Calcium Phosphate Transfection Kits (631312, Clontech) or PolyFect Reagent (301107, Qiagen), following the manufacturer's instructions. None of the cell lines used in this study was found in the database of commonly misidentified cell lines that is maintained by ICLAC and NCBI Biosample. All cell lines were tested and confirmed to be free of mycoplasma.

For bone marrow-derived macrophages (BMDMs) isolation and culture[69], bone marrow collected from mouse tibias and femurs was plated on sterile petri dishes and was incubated for 7 days in DMEM containing 10% Tet system approved FBS, 1% penicillin-streptomycin, 2 mM L-glutamine, and 20% (vol/vol) conditioned medium from L929 mouse fibroblasts. To induce UVRAG$^{FS}$ expression, cells were treated with 1 µg/mL of Dox on the 3$^{rd}$ day after plating. For inflammation studies, BMDMs were incubated with LPS (100 ng/ml) for 6 h and then were treated with ATP (1 mM) for 1 h[4], or transfected with vehicle or poly(dA:dT) (1 µg/ml) for 6 h using LyoVec™ (lyec-2, InvivoGen). For inflammation inhibition, BMDMs were primed with 100 ng/ml of LPS for 6 h followed by treatment with 10 µM Z-YVAD-FMK (ALX-260-154-R020, Enzo Life Sciences) or 1 µM of MitoQ (10-1363-0005, Focus Biomolecules) for 1 h and followed by stimulation with 1 mM ATP treatment for 1 h.

**Plasmid constructs**. The mRFP-EGFP-LC3 plasmid and p40(phox)-PX-EGFP plasmid were kindly provided by Drs. Jae U Jung (University of Southern California) and S. Field (University of California, San Diego), respectively. All constructs were confirmed by sequencing using an ABI PRISM 377 automatic DNA sequencer (Applied Biosystems).

**Antibodies, fluorescent dyes, and other reagents**. The following antibodies were used in this study: polyclonal rabbit anti-UVRAG (C-term) (AP1850b, Abgent, 1:1000 for WB, 1:200 for IP), polyclonal rabbit anti-LC3B (2775S, Cell Signaling Technology, 1:1000 for WB), polyclonal rabbit anti-p62 (5114S, Cell Signaling Technology, 1:1000 for WB), polyclonal rabbit anti-p62 (18420-1-AP, Proteintech, 1:500 for IHC), monoclonal mouse anti-p62 (814802, Biolegend, 1: 50 for IF), monoclonal rabbit anti-Atg16 (8089T, Cell Signaling Technology, 1:1000 for WB), monoclonal rabbit anti-Atg5 (GTX62601, GeneTex, 1:1000 for WB), polyclonal rabbit anti-Beclin-1 (11306-1-AP, Proteintech, 1:1000 for WB, 1:200 for IP), polyclonal rabbit anti-PI3KC3 (AP8014a, Abgent, 1: 1000 for WB), polyclonal rabbit anti-Rubicon (GTX129096, GeneTex, 1:1000 for WB), polyclonal rabbit anti-p-UVRAG (Ser498) (ABS1600, EMD-Millipore, 1:1000 for WB), monoclonal rabbit anti-MyD88 (4283, Cell Signaling Technology, 1:1000 for WB), polyclonal rabbit anti-TRAF6 (PA5-29622, Invitrogen, 1:1000 for WB), polyclonal rabbit anti-TRIF (GTX13810, GeneTex, 1:1000 for WB), monoclonal rabbit anti-Ubiquitin (linkage-specific K63) (ab179434, Abcam, 1:1000 for WB), polyclonal rabbit anti-Bcl-2 (2876S, Cell Signaling Technology, 1:1000 for WB), monoclonal rabbit anti-Caspase-1 (ab108362, Abcam, 1:1000 for WB, 1:200 for IP), polyclonal goat anti-IL-1β (AF-401-SP, R&D Systems, 1:1000 for WB), polyclonal rabbit anti-IL-18 (ab191860, Abcam, 1:1000 for WB), polyclonal rabbit anti-IL-18 (210-401-323, Rockland, 1:1000 for WB), monoclonal mouse anti-Caspase-11 (sc-374615, Santa Cruz, 1: 1000 for WB), monoclonal rabbit anti-NLRP3 (15101S, Cell Signaling Technology, 1:1000 for WB), polyclonal rabbit anti-NLRC4 (PA5-79739, Invitrogen, 1:1000 for WB), polyclonal rabbit anti-NLRP6 (A15628, ABclonal, 1:1000 for WB), polyclonal rabbit anti-AIM2 (63660s, Cell Signaling Technology, 1:1000 for WB), monoclonal rabbit anti-ASC (67824T, Cell Signaling Technology, 1:1000 for WB), monoclonal rabbit anti-Axin1 (2087s, Cell Signaling Technology, 1:1000 for WB), polyclonal rabbit anti-APC (NBP2-15422, Novus Biologicals, 1:1000 for WB), monoclonal mouse anti-Parkin (PRK8) (sc-32282, Santa Cruz Biotechnology, 1:1000 for WB, 1:200 for IF), monoclonal rabbit anti-Tom20 (42406S, Cell Signaling Technology, 1:1000 for WB, 1:200 for IP, 1:200 for IF), monoclonal rabbit anti-p-IκBα (2859T, Cell Signaling Technology, 1:1000 for WB), monoclonal mouse anti-IκBα (4814T, Cell Signaling Technology, 1:1000 for WB), monoclonal rabbit anti-NFκB (8242T, Cell Signaling Technology, 1:400 for IF), polyclonal rabbit anti-Ki67 (NB110-89719SS, Novus Biologicals, 1:500 for IHC), polyclonal rabbit anti-cleaved caspase-3 (9661T, Cell Signaling Technology, 1:200 for IHC), polyclonal rabbit anti-γ-H2AX (NB100-384, Novus Biologicals, 1:3000 for WB), monoclonal rabbit anti-Keratin 20 (13063T, Cell Signaling Technology, 1:800 for IHC), polyclonal rabbit anti-E-cadherin (20874-1-AP, Proteintech, 1:200 for IHC and 1:1000 for WB), polyclonal rabbit anti-N-cadherin (PA5-29570, Thermo Fisher, 1:1000 for IHC and 1:1000 for WB), monoclonal rabbit anti-Vimentin (5741T, Cell Signaling Technology, 1:1000 for IHC & WB), monoclonal rabbit anti-GSK3β (12456T, Cell Signaling Technology, 1:1000 for WB), monoclonal rabbit anti-p-GSK3β (Ser9) (5558T, Cell Signaling Technology, 1:1000 for WB), monoclonal mouse anti-β-catenin (MA1-300, Thermo Fisher, 1:1000 for WB), monoclonal rabbit anti-phosphor-β-catenin (Ser33/37/Thr41) (8814S, Cell Signaling Technology, 1:500 for IHC, 1:1000 for WB), monoclonal rabbit anti-Cyclin D1(2978T, Cell Signaling Technology, 1:1000 for WB), monoclonal mouse anti-c-Myc (626802, Biolegend, 1:1000 for WB), polyclonal rabbit anti-c-Myc (10828-1-AP, Proteintech, 1:400 for IHC), polyclonal rabbit anti-CD20 (PA5-16701, Thermo Fisher,

1:50 for IHC), polyclonal rabbit anti-Bcl-6 (21187-1-AP, Proteintech, 1:500 for IHC), polyclonal rabbit anti-Kappa (14678-1-AP, Proteintech, 1:400 for IHC), polyclonal rabbit anti-Cytokeratin 7 (NBP1-88080, Novus Biologicals, 1:500 for IHC), monoclonal rabbit anti-TTF-1 (ab76013, Abcam, 1:300 for IHC), monoclonal mouse anti-TCF4 (GTX52873, GeneTex, 1: 1000 for WB), monoclonal mouse anti-Flag (F7425, Sigma, 1:1000 for IHC & WB), and polyclonal rabbit anti-γ-Tubulin (GTX113286, GeneTex, 1: 1000 for IF). HRP-labeled or fluorescently labeled secondary antibody conjugates, purchased from Molecular Probes (Invitrogen). Purified rabbit IgG was purchased from Pierce. Z-YVAD-fmk was from Enzo Life Sciences (ALX-260-154-R020); Mitoquinone (MitoQ) was from Focus Biomolecules (10-1363-0005); Torin1 was from Selleckchem; 3-MA was from Santa Cruz Biotechnology; MCC950 was from Adipogen; Dextran Sulfate Sodium (DSS) was purchased from Affymetrix; Doxycycline, azoxymethane (AOM), ATP, Oil Red O, and LPS (*Escherichia coli*), chloroquine (CQ) were from Sigma. Unless otherwise stated, all chemicals were purchased from Sigma.

**Autophagy analyses**. For assessment of autophagy in vivo, 6–8-week-old Dox-treated *control;GFP-LC3* or *iUVRAG$^{FS}$;GFP-LC3* mice were either subjected to starvation for 48 h or challenged with 20 mg kg$^{-1}$ LPS from *Escherichia coli* intraperitoneally. Mice were then perfused with 4% paraformaldehyde (PFA) in PBS and tissues were collected and processed for frozen sectioning[21]. The total number of GFP-LC3 puncta was counted per 2620 µm$^2$ area (20 randomly chosen fields were used per mouse) or per colon crypt structure and the average value for each tissue for each mouse was determined by two independent researcher blinded to genotype. The mouse skeletal muscle, heart, liver, and colon tissue sections were imaged using a ×60 Nikon objective (PL APO, 1.4 NA).

For western blot analysis, frozen tissues were lysed in ice-cold RIPA lysis buffer (50 mM Tris-HCl pH 8.0, 150 mM NaCl, 1 mM EDTA, 1% Triton X-100, 0.5% sodium deoxycholate, 0.1% SDS) containing protease inhibitor cocktail (A32953, Pierce) for 30 min at 4 °C. Lysates were centrifuged at 16,000 × g for 10 min at 4 °C. Cleared lysates were diluted in 2X SDS-PAGE loading buffer and analyses using antibodies as indicated.

**Immunofluorescence and confocal microscopy**. Cells plated on coverslips were fixed with 4% paraformaldehyde (30 min at RT). After fixation, cells were permeabilized with 0.2% Triton X-100 for 5 min and blocked with 10% goat serum (G9032, Sigma) for 2 h at RT. Primary antibody staining was carried out using antiserum or purified antibody in 1% goat serum for 1–2 h at RT or overnight at 4 °C. Cells were then extensively washed with PBS and incubated with diluted Alexa 488-, Alexa 594-, and/or Alexa 633-conjugated secondary antibodies in 1% goat serum for 1 h, followed by DAPI (4′, 6′-diamidino-2-phenylindole) staining. Cells were mounted using Vectashield (Vector Laboratories, Inc.). Confocal images were acquired using a Nikon Eclipse C1 laser-scanning microscope (Nikon, PA), fitted with a 60× Nikon objective (PL APO, 1.4NA), and Nikon imaging software. Images were collected at 512 × 512 pixel resolution. The stained cells were optically sectioned in the z-axis. For multichannel imaging, fluorescent staining was imaged sequentially in line-interlace modes to eliminate crosstalk between the channels. The step size in the z-axis varied from 0.2 to 0.5 mm to obtain 16 slices/imaged file.

For immunofluorescence (IF) in mouse tissues (spleen, lung, colon, and liver)[70], samples were harvested and fixed in 10% neutral buffered formalin and embedded in paraffin. Tissues were cut into 6 µm sections and placed on microscope slides (Fisher Scientific). For staining, slides were deparaffinized in xylene and rehydrated in alcohol. Slides were washed with 0.1 M glycine in PBS (IF wash). Antigen retrieval was performed by incubating the sections in boiling 10 mM sodium citrate pH 6.0. Slides were washed 3 times with the IF wash followed by incubation with blocking solution (10% goat serum in IF wash) for 1 h at room temperature. Slides were incubated overnight at 4 °C in primary antibody, then washed three times with IF wash and incubated in Alexa 594-conjugated secondary antibody for 1 h in the dark. Afterward, slides were washed 3 times with IF wash and mounted with ProLong Gold Antifade containing DAPI (4′,6-diamidino-2-phenylindole) (Invitrogen).

For image quantification, ~50–100 cells, randomly chosen from 10 high power fields and pooled from three independent experiments, were evaluated for the distribution pattern of the indicated molecules. The Pearson correlation coefficient was calculated using the built-in colocalization analysis module of the NIS-Elements AR software. All experiments were independently repeated several times. The investigators conducted blind counting for each quantification-related study.

**Immunohistochemistry**. Tissue sections were fixed in 10% neutral buffered formalin and embedded in paraffin. Tissue sections were routinely stained with hematoxylin and eosin. For immunohistochemistry staining, tissue slides were deparaffinized in xylene and rehydrated in alcohol. Endogenous peroxidase was blocked with 3% hydrogen peroxide. Antigen retrieval was performed by incubating the sections in boiling 10 mM sodium citrate pH 6.0 followed by incubation with the indicated primary antibody overnight at 4 °C. Antibody binding was detected with EnVision™ Dual Link System-HRP DAB kit (K4010, Dako). Sections were then counterstained with hematoxylin. For negative controls, the primary antibody was excluded. For evaluation and scoring of immunohistochemical data, we randomly selected 10 fields within the tumor area under high power

magnification (×40) for evaluation. The investigators conducted blind counting for all quantification.

**Histopathology analysis of colon**. Histopathology assessment was performed by an anatomical pathologist (A. H) from de-identified section slides[71]. The features evaluated covered: acute and/or chronic inflammation, hyperplastic changes of the colon epithelium, and crypt distortion or damage, fibrosis, and neoplasia. Three independent parameters were measured: inflammatory cell infiltrate, extent of hyperplasia, and crypt damage. The total histological score was calculated by summing of the three independent scores with a maximum score of 12. Inflammation was assessed using a scoring system from the literature[72,73]: $0 =$ no inflammation; $1 =$ mild chronic mucosal inflammation; $2 =$ mild acute or moderate chronic mucosal and submucosal inflammation; $3 =$ severe acute or chronic mucosal, submucosal, and transmural inflammation. Hyperplastic changes were scored as the increase in epithelial cell numbers in crypts relative to baseline epithelial numbers per crypt as follows: $0 =$ none or minimal (<20%); $1 =$ mild (21–35%); $2 =$ moderate (36–50%); $3 =$ marked (>50%). Crypt damage was scored as $0 =$ none; $1 =$ only surface epithelium damaged; $2 =$ surface crypt and epithelium damaged; $3 =$ entire crypt lost and surface epithelium damaged; $4 =$ entire crypt and epithelium lost.

**Conventional electron microscopy**. Mice were euthanized, and liver was rapidly fixed overnight at 4 °C in 1/2 strength Karnovsky's (2% paraformaldehyde and 2.5% glutaraldehyde in 0.2 M sodium cacodylate buffer, pH 7.4). Tissues were post-fixed in 2% osmium tetroxide in 200 mM sodium cacodylate for 2 h at 4 °C and rinsed in 0.1 M cacodylate buffer. Samples were then blocked, and stained with 1% uranyl acetate overnight at 4 °C. The pellet was then rinsed with 0.1 M sodium acetate. Samples were dehydrated through a graded series of ethanol, and then infiltrated with Epon resin overnight at room temperature. They were then embedded in resin overnight at 60 °C. Thin sections were cut on a Leica Ultracut R, and collected onto formvar-carbon coated slot grids. Sections were examined on a JEOL 2100 transmission electron microscope. Images were recorded on film at ×5000 magnification.

**Metaphase spreads**. For metaphase spreads of splenocytes[70], freshly harvested spleens were minced and filtered through a 40 µm cell strainer into warm PBS. Cells were spun at $1000 \times g$ for 5 min, resuspended in warm DMEM supplemented with 10% FBS, 100 U/mL penicillin,100 U/mL streptomycin. Cells were then treated with colcemid (KaryoMAX, GIBCO) at 0.1 µg/ml for 1 h at 37 °C[74]. Cells were swollen in prewarmed 75 mM KCl for 30 min at 37 °C, then carefully fixed in methanol: acetic acid (3:1) and kept at −20 °C. Metaphase spreads were prepared by dropping cells in the fixative onto Superfrost glass slides (Fisher Scientific) at 25 °C and 60% of humidity. After air dry, metaphase images were captured and analyzed using a 20X Nikon objective (PL APO, 1.4 NA). At least 10 metaphases from each tissue were scored for chromosomal aberration.

**Immunoblotting and immunoprecipitation**. For co-immunoprecipitation from mice tissues, frozen tissues were weighed and homogenized in ice-cold lysis buffer (25 mM HEPES, 150 mM NaCl, 1 mM EDTA, 1% Triton X-100; 1 ml per 100 mg tissue) containing a complete protease inhibitor cocktail (A32953, Pierce). Lysates were centrifuged ($16,000 \times g$ at 4 °C for 30 min) and the supernatant were pre-cleared with protein A/G agarose beads for 2 h at 4 °C. Lysates were used for immunoprecipitation (IP) with the indicated antibodies. Generally, 1–4 µg commercial antibody was added to 1 ml lysates and incubated at 4 °C for 8–12 h. After addition of protein A/G agarose beads, incubation was continued for another 2 h. Immunoprecipitates were extensively washed with IP wash buffer (10 mM Tris at pH 7.5, 150 mM NaCl, 1 mM EDTA, 0.2% Triton X-100) supplemented with protease inhibitor cocktail and then eluted with SDS-PAGE loading buffer by boiling for 5 min.

For co-immunoprecipitation from cell culture, cells were washed with ice-cold PBS, lysed in 2% Triton X-100 lysis buffer (20 mM Tris at pH 7.5, 150 mM NaCl, 1 mM EDTA and 2% Triton X-100) supplemented with protease inhibitor cocktail before co-immunoprecipitation using the same protocol described for mouse tissues.

For immunoblotting, eluates were resolved by SDS-PAGE and transferred to a PVDF membrane (BioRad). Membranes were blocked with 5% non-fat milk or BSA, and probed with the indicated antibodies. Horseradish peroxidase (HRP)-conjugated goat secondary antibodies were used (1:3000, Invitrogen). Immunodetection was achieved with Hyglo chemiluminescence reagent (Denville Scientific), and detected by ChemiDoc Imaging System (Bio-Rad).

**Enzyme-linked immunosorbent assay**. Mouse cytokines in serum or culture supernatants were measured with Enzyme-linked immunosorbent assay (ELISA) kits (R&D Systems) for IL-1β (MLB00C), IL-6 (M6000B), TNF-α (MTA00B), and IFN-β (MIFNB0), according to the manufacturer's directions. Mouse IL-18 was measured by ELISA (7625, MBL international).

For cytokine levels in the colon extracts[75], colon tissues were homogenized with RIPA buffer containing protease inhibitor cocktail. The supernatant was collected and tested for IL-1β, IL-6, and TNF-α according to the manufacturer's directions.

**Bacteria colony forming units (c.f.u.)**. Levels of c.f.u. in freshly isolated mice feces were determined by homogenization of feces in 0.01% Triton X-100 in PBS followed by serial dilution plating on non-selective Luria-Bertani agar plates.

**Cell viability**. Cell viability was determined using the CellTiter 96® non-radioactive cell proliferation assay (G4001, Promega) following the manufacturer's instructions.

**Cell cycle analysis**. Cell cycle analysis was determined by flow cytometry (BD Biosciences, Franklin Lakes, NJ, USA). SW480 cells were harvested, washed with phosphate-buffered saline (PBS), fixed with 70% cold ethanol overnight at 4 °C, and then incubated with a staining solution containing 100 µg/ml RNase and 50 µg/ml propidium iodide (PI; ab139418, Abcam) for 1 h at 37 °C before analysis by flow cytometry. A total of $10^4$ nuclei were examined by a BD FACSCanto II flow cytometer. For analysis, first gate on the single cell population using pulse height vs. pulse area. Then apply this gate to the scatter plot and gate out debris. Gates are combined and applied to the PI histogram plot. DNA histograms were analyzed by FlowJo software (both from Becton-Dickinson, Mountain View, CA, USA). Results are presented as the percentage of cells in each phase.

**Measurement of mitochondrial membrane potential**. TMRM (AS-88065, AnaSpec) was used to measure mitochondrial membrane potential (Δψm). Briefly, Dox (1 µg/ml)-treated BMDMs were primed with LPS (100 ng/ml) for 6 h and stimulated with ATP (1 mM) for 1 h. The cells were then washed twice with PBS and incubated with 200 nM of TMRM and 25 nM of MitoTracker Green for 30 min at 37 °C and washed twice with PBS. The cells were resuspended in PBS and fluorescence intensity was measured with a FLUOstar Omega microplate reader (BMG Labtech). The ratio of TMRM to MTG fluorescence intensity was normalized to PBS controls.

**Mitochondrial ROS detection**. MitoSOX (M36008, Invitrogen) was used to measure mitochondrial ROS production. Briefly, Dox (1 µg/ml)-treated BMDMs were primed with LPS (100 ng/ml) for 2 h and stimulated with ATP (1 mM) for 30 mins. The cells were then washed twice with PBS and incubated with 5 µM of MitoSOX for 10 min and washed twice with PBS again. Fluorescence intensity was determined using a Synergy 2 multimode plate reader (BioTek Instruments), and the data were normalized to PBS controls.

**Measurement of cytosolic mitochondrial DNA (mtDNA)**. Dox (1 µg/ml)-treated BMDM were primed with LPS (100 ng/ml) for 2 h and stimulated with ATP (1 mM) for 30 mins. Cytosolic mtDNA was measured using a mitochondrial isolation kit (89874, ThermoScientific) according to manufacturer's instructions. Mitochondrial DNA encoding cytochrome c oxidase 1 (COX1) was measured by quantitative qPCR (refer to Supplementary Table 2 for primers' sequence) with same volume of the DNA solution. Nuclear DNA encoding 18S ribosomal RNA was used for normalization.

**Clonogenic cell survival assay**. The log-phase cells were trypsinized, counted, and plated at appropriate dilutions in 6-well plates for colony formation. After 14 days of incubation in the presence/absence of iCRT14 (SML0203, Sigma) treatment (50 uM), colonies were fixed and stained with colony fixation-staining solution containing glutaraldehyde 6.0% (vol/vol) and crystal violet 0.5% (wt/vol) in H2O, and counted. Plating efficiency (PE) was determined for each individual cell line as described[76], and the surviving fraction (SF) was calculated based on the number of colonies that arose after treatment, expressed in terms of PE. Each experiment was repeated six times.

**Gene knockdown by shRNA and lentiviral gene delivery**. All shRNAs were purchased from Open Biosystem. Lentiviral-compatible shRNAs against Beclin1 (sh1: V2LHS_241693, sense: TGTTGGTCATCTCCAGGCG; sh2: V3LHS_332992, sense: TCGCTAGGCAGCTCCTGCT), UVRAG (sh1: V2LHS_197759, sense: ATTGTAACTGGACTCCAGG; sh2: V3LHS_357540, sense: ATGACATCATCA ATCTCCT). For lentivirus production, HEK293T cells were transfected with the transfer vector (e.g. pCDH-CMV-MCS-EF1-Puro or pGIPZ), pCMV-dR8.91 packaging plasmid, and pCMV-VSV-G envelope plasmid in a 5:1:4 ratio using the Calcium Phosphate Transfection Kit (Clontech). The medium was replaced 12 h later. Viral particles were collected 48 h post-transfection, filtered with 0.45 µm sterile filter, and concentrated overnight by Lenti-X concentrator (631312, Takara) at a ratio of 3:1, followed by centrifugation at 4 °C ($28,800 \times g$, 2 h, ThermoFisher Sorval RC 6+). Viral particles were re-suspended in fresh medium with 8 µM/mL polybrene, and were plated with target cells for 24 h. Lentiviral-transduced cells were selected in 2 µg/mL puromycin for 7 days with the medium changed daily.

**RNA extraction, cDNA synthesis, and qPCR Analysis**. Total RNA was extracted from mice tissue or cell culture using TRIzol (15596-026, Invitrogen) and purified with RNeasy Plus Mini Kit (Qiagen 74104), following the manufacturer's instructions. In all, 1 µg of total RNA was used for cDNA synthesis using iScript™ cDNA

Synthesis Kit (1708891, Bio-rad). Quantitative real-time PCRs were carried out using the primers listed in Supplementary Table 2 and iQ SYBR Green Master Mix (Bio-rad). Samples were obtained and analyzed on the CFX96 Touch Real-Time PCR Detection System (Bio-Rad). The gene expression levels were normalized to actin.

**CNV analysis**. Genomic DNA was isolated from bulk sample with TRI Reagent BD from Sigma-Aldrich (T3809). Concentration of DNA was quantified with Qubit Fluorometric Quantification (Thermo Fisher). Amplified DNA was sheared using sonication (Covaris S2/E210 focused-Ultrasonicater) with the microtube setup and the 200 bp target size protocol for DNA shearing. In all, 30 ng of sonicated DNA was used for library construction using the NEBNext Ultra DNA Library Preparation Kit for Illumina (New England Biolabs, Cat#. E7370L). The constructed library DNA concentration was quantified with Qubit (Thermo Fisher) and the expected library size distribution of 300–500 bp was confirmed using the Agilent 2100 Bioanalyzer (High-Sensitivity DNA Assay and Kit, Agilent Technologies, Cat#. 5067–4626). The individual libraries from barcoded samples were pooled. The pooled libraries were cleaned using AMPure XP Beads (Beckman Coulter Inc., Cat# A63882). Libraries were sequenced using the Illumina NextSeq 500 or the HiSeq2500 SR50 generating fastq files. In all, 30 bp was trimmed off the 5′ end of each read to remove the WGA4 adapter sequence before alignment to the hg19 reference genome using the Bowtie algorithm. The resulting BAM file was sorted and PCR duplicates were removed using SAMtools. The number of reads falling into each of 5000 'bins' comprising the entire UCSC reference genome, was calculated using a previously published Python script[61]. Finally, an R script utilizing the Bioconductor package, DNAcopy_1.26.0 (http://bioconductor.org/packages/DNAcopy/), was used to normalize and segment the bin counts across each chromosome generating a genome-wide CNV profile.

**Statistical analysis**. Statistical significance was performed using unpaired Student's t-test, One-way analysis of variance (ANOVA) with Tukey's test to correct for multiple comparisons or Two-way ANOVA with multiple comparisons test unless otherwise stated. Log-rank analysis was used for survival curve. All statistical analyses were assessed using GraphPad Prism 7.0 (GraphPad Software, Inc.), unless otherwise stated. Data are presented as the mean ± SD. A $p$ value of <0.05 was considered statistically significant. All experiments were independently repeated at least three times. To ensure adequate power and decrease estimation error, we used large sample sizes and multiple independent repeats by independent investigators. Multiple lines of experiments including different quantification methods were used for consistent and mutually supportive results. The sample size was chosen according to well-established rule in the literature as well as our ample experience in previous research.

**Reporting summary**. Further information on research design is available in the Nature Research Reporting Summary linked to this article.

## Data availability
Generated mouse model and cell lines are available from the corresponding author upon request. All other data that support the findings of this study are available from the corresponding author on reasonable request. The source data underlying Figs. 1a–c, 2a–c, 3a, c, d, f–i, 4b, c, e, g–j, l, 5b–d, g, 7b, d, f, h, k–m, and 8a, d, and Supplementary Figs. 1b, 2b, 3b, d–k, 4a, c, d, h, n, 5c, 7i, j, and 8a are provided as a Source Data file.

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

## Acknowledgements

We thank Drs. J.U. Jung, P. Feng, N. Mizushima, and Y. Ohsumi for providing critical reagents, Dr. A. Rodriguez for expert technical assistance in electron microscopy, all members of the C.L. laboratory for helpful discussion. We acknowledge the financial support from China Scholarship Council (CSC) of H.G. for his PhD study at University of Southern California (USC). This work was supported by NIH grants R01CA140964 (to C.L.) and R01ES029092 (to C.L.).

## Author contributions

C.Q., Y.S., and H.G. performed research and analyzed the data. C.Q. conducted most mouse-related experiments and pathology analysis; Y.S. and H.G. conducted most biochemistry and molecular biology experiments. S.L., H.M., and N.S. helped with bone marrow chimera experiments. M.F., D.C., B.C., D.N., and N.W. contributed to the transgenic animals construction and characterization. S.R. and J.H. guided CNV analysis of mice tumors. S.E.M. and A.H. helped with IHC and pathological analyses. V.P., O.A., G.I., S.M.M., and H.C. participated in discussion. C.L. designed research, analyzed data, and wrote the paper. All authors read, contributed to, and approved the final manuscript.

## Competing interests

The authors declare no competing interests.
