## [Peer Review File · Nature Communications]

Reviewers' comments:

Reviewer #1, Expertise: Autophagy, stress (Remarks to the Author):

Christine Quach et al, Nature Communication 2019

Minor Revision

The work by Quach et al. is very extensive in vivo work demonstrating that UVRAG truncated mutation (UVRAGFS) affects inflammation and tumorigenesis through the regulation of autophagy. From the generation of a transgenic inducible mouse model expressing UVRAG frame-shift mutation (iUVRAGFS) to the dissection of the in vivo role of UVRAGFS, the authors demonstrated that this mutation results in: i) impaired starvation/endotoxin-induced autophagy activation, ii) increased inflammatory response (through NLRP3 inflammasome activation and IL-1 β secretion) and associated pathologies, iii) increased age-related spontaneous malignancies. This last phenotype has been linked to the capability of UVRAGFS to ensure beta-catenin stabilization/activation by inhibiting its autophagic degradation.

Overall the manuscript is very well structured and the finding supported by the data; each analysis has been extremely well-performed, with very detailed and meticulous approaches.

However, a few major concerns remain:

Major comments:

- The authors completely disregard the UVRAGFS role in DNA damage repair and chromosomal instability/centrosome amplification (that was previously published by the same authors, ref. 20) as a possible additional contribution to the severe phenotypes so far observed, especially in relation to the tumorigenic potential of this mutation. This further, very relevant, function of the truncated mutation of UVRAG has not been explored in the present work. Therefore, it is highly recommended to include this part too, particularly in relation to the role of UVRAGFS in promoting spontaneous tumorigenesis (last paragraphs in the Results section).
- The actual tumorigenic power of the FS mutation should be further validated and supported. To do so, the authors are highly recommended to investigate the key findings upon UVRAG depletion at least in vitro, or in a KO conditional murine model.

I also have some additional:

Minor comments:

- The characterization of the conditional Flag/tagged UVRAGFS-luciferase transgene expression was carried out in a very precise and convincing fashion. However, the authors should provide the concept, for which the expression was checked only in spleen and colon. Indeed, part of the following analyses were carried out also in other organs (refer to Fig. 1a, S1a and j).
- Please, provide a WB analysis for colon in Figure S1c.
- It would be interesting to also investigate on the effects of LDs accumulation observed in livers of starved iUVRAGFS Dox-treated mice. Did the authors observe any lipid-related toxicity in hepatocytes?
- Can LD accumulation also reflect an increase in lipids uptake/biosynthesis or lipolysis deregulation beyond autophagic-mediated LDs clearance? Could this accumulation boost liver pathologies such as fatty liver disease and HCC? The authors should address these points in the discussion.
- "[...] showed massive enlargement and accumulation of lipid droplets (LD) in liver when compared to control mice (Fig. 1c)". Please, also refer to Fig. S1j.
- In Fig. 1b, make the statistics between Ctr+Dox and iUVRAGFS+Dox, according to what it is shown for panel a.
- Negative controls should be included in the Co-immunoprecipitation analyses (e.g. Fig. 1d, 2b, Fig. S2b).

- The authors should address why increased levels of caspase 1 and of IL-1beta do not match with reduction in the levels of the pro-caspase 1 and -IL-1beta, respectively.
- Fig. 2b, its interpretation in the results section (p. 8) is misleading: "Co- immunoprecipitation analysis showed that LPS treatment resulted in increased interaction...ubiquitination"; indeed, no interaction/ubiquitination is shown in the PBS-treated samples (left side of the image). Rephrase the sentence accordingly.
- In relation to mtROS production, the authors should also assess $\Delta\Psi_m$ and mitophagy markers.
- Could mtROS production be linked to a putative (increased) DNA damage?
- Please, analyse chemokines expression also in Fig. 3.
- In relation to Fig. 4d, it is mentioned in p.12 that "enhanced neutrophil infiltration" occurs in the tissue analysed; this is not clear from the image, a higher magnification and/or arrows should be included.
- Fig. 4l, add the blue/red legend as in Fig. 4i.
- Together with NLRP3 inhibition by MCC950, authors should also include inhibition of caspase 1 and eventually assess whether one of the two proteins has more relevance in the phenomena observed.
- Fig. 5a is not clear. Please, provide a higher magnification or crop/zoom in the relevant areas.
- The accumulation of p62 in Fig. 5h is not convincing.
- In relation to Fig. 7: did the authors also observed an actual effect of iUVRAGFS on the cell cycle?
- In Fig. 6a, 7 and Suppl. Fig. 6a is not clear how many mice have been analysed per each group.
- The authors should assess whether the EM genes are actually expressed in a β -catenin dependent manner.
- It is recommended to confirm the role of β -catenin by its inhibition at least in vitro.
- Add reference number in the Mat & Met section on Bone marrow-derived macrophages (BMDMs) isolation and culture.

Reviewer #2, Expertise: CRC, models, Wnt(Remarks to the Author):

This manuscript is generated by a team with over a decade of expertise examining autophagy and the UVRAG protein in multiple contexts. In this study, they noted that a truncating mutation resulting from a frameshift (UVRAGFS), leads to a phenotype in which autophagy is severely impaired. Drawing on their recent publication in which this mutation increases the oncogenic properties and chemosensitivity in colorectal cancer, they extend these observations by creating a doxycycline-inducible murine model expressing this protein. They then examine the impact on models of colitis and colitis-associated cancer with traditional biochemical and cancer biology approaches.

Overall, the premise for the experiments and the findings are clear. The authors and co-authors should be commended on adding more depth to the understanding, mechanisms, and models in the area of autophagy. However, there are areas where the data are not clear and therefore the interpretations are likewise less clear.

1. The interactions of the IP with Beclin1 then IB with Bcl2 and UbK63 are a bit difficult to discern. Is there a way to quantify these findings?
2. The data in Figure 7k, regarding p62 are not convincing.
3. Since the Rosa26 transgenic expression is global, why were only BMDM studied?
4. Was there an attempt to create an inducible UVRAGFS murine model with expression only in the colon, or in the small bowel and proximal colon?
5. Were other models of enteritis or colitis such as Samp-Yit or Winnie mice explored?
6. Were other acute or chronic chemokine/cytokine expression noted in the KC, MIP2, CXCL1 families?
7. In Figure 5e, was only a single mouse used? Is there quantification of the E-cadherin and N-cadherin?

8. What are the thoughts regarding the localization and types of malignancies that spontaneously developed (DLBCL)?

9. Is there an area in the supplementary materials where the entire western blots are available rather than cropped versions?

Emina Huang

Reviewer #3, Expertise: CRC, inflammation (Remarks to the Author):

Manuscript Title: "A truncating mutation in the autophagy gene UVRAG drives inflammation and tumorigenesis in mice"

In this manuscript, the authors generated an elegant inducible mouse model that expresses a truncated dominant negative form of UVRAG (a form expressed in several cancers; denoted iUVRAGFS) to address the role of UVRAG in autophagy *in vivo* and of the impact of its deregulation in inflammatory diseases and cancer. They demonstrated that iUVRAGFS impaired starvation-induced autophagy by inhibiting PIK3C3 formation. In LPS-induced autophagy, iUVRAGFS inhibited Beclin-1 K63 Ub by TRAF6. The authors then went on to study the response of these mice to experimental models of inflammatory disease, namely sepsis and colitis and to explore the role of autophagy in these inflammatory conditions. In LPS endotoxemia, they showed exacerbation of the disease, which was associated with excessive production of the inflammasome-dependent cytokines IL-1b and IL-18. In parallel, they showed that the NLRP3 inflammasome pathway is excessively activated by LPS+ATP in BMDM *ex vivo*. In DSS-induced colitis, iUVRAGFS expressing mice were also more susceptible than wild-type mice and this phenotype was alleviated with NLRP3 pharmacological inhibition. Last, aged iUVRAGFS mice developed spontaneous B cell lymphomas, lung adenomas, and a hyperplastic response in the colon.

This is a very interesting paper; the experiments are well performed and the results are convincing. However, there are several loose ends that need to be tightened prior to publication. By presenting too many models, the authors reinforce the role of UVRAG in autophagy but unfortunately end up with less profound analysis of each model.

1. In the LPS endotoxemia model, caspase-11 is the key inflammasome involved in IL-1b and IL-18 production. The authors need to show caspase-11 levels and activation. They presume based on the BMDM *ex vivo* results that NLRP3 is involved in mediating the hyper-inflammatory response in this endotoxemia model. This needs to be demonstrated. The authors can cross their mice to NLRP3-deficient mice, or as they did in the DSS colitis model, they can use the NLRP3 inhibitor MCC-950.

2. It has been recently reported that not only NLRP3 but also AIM2 inflammasome is regulated by autophagy (and not mitophagy) through control of ASC levels. Therefore, ASC levels should be shown in the different models. Also, it will be interesting to examine the response of BMDM to polydA:dT to assess the AIM2 inflammasome.

3. In the DSS colitis model, IL-18 production is essential in tissue repair. Further, different inflammasomes are activated in this model including NLRP6, NLRC4, AIM2 (all 3 in the epithelium) and NLRP3 (in myeloid cells). It will be important to show levels and activation of these inflammasomes and levels of IL-18.

4. What is the autophagy target in B cell lymphoma? And lung adenoma? The authors should discuss these findings.

5. In the colon hyperplastic response, while the authors excluded NLRP3 contribution, they haven't excluded a role of other inflammasomes (that could be hyperactivated due to ASC accumulation), and of IL-18!! This is important as this cytokine is involved in IEC proliferation.

RESPONSE TO REVIEWERS

We are truly appreciative of all the reviewer's constructive and insightful comments, according to which the manuscript has been rigorously and substantially revised. We hope the new version of our manuscript is now appropriately suited for publication in *Nature Communications*. A detailed response to the Reviewer's critiques and a description of the new experiments (in *italic*) follow:

Point-by-point Response to Reviewer #1

Reviewer 1 commented that "The work is very extensive in vivo work demonstrating that UVRAG truncated mutation (UVRAG^{FS}) affects inflammation and tumorigenesis through the regulation of autophagy. The manuscript is very well structured and the finding supported by the data; each analysis has been extremely well-performed, with very detailed and meticulous approaches".

Response: *We greatly appreciate the reviewer's enthusiasm and positive comments. Rigorous efforts have been taken to address the deficits noted by the Reviewer, as detailed below.*

Major comments:

1) The authors completely disregard the UVRAG^{FS} role in DNA damage repair and chromosomal instability/centrosome amplification (that was previously published by the same authors, ref. 20) as a possible additional contribution to the severe phenotypes so far observed, especially in relation to the tumorigenic potential of this mutation. This further, very relevant, function of the truncated mutation of UVRAG has not been explored in the present work. Therefore, it is highly recommended to include this part too, particularly in relation to the role of UVRAG^{FS} in promoting spontaneous tumorigenesis (last paragraphs in the Results section).

Response: *We thank the reviewer for this insightful comment. As noted by the reviewer, the role of UVRAG^{FS} in cancer is also linked to its ability to promote centrosome amplification and thereof chromosomal instability in vitro (He et al., 2015). To address this, we further investigated the effect of UVRAG^{FS} on centrosome homeostasis in vivo, which led to a brand new dataset of Fig. 8 (also attached herein for Reviewer's reference) and Fig. S8. Specifically, we observed a significant increase in the incidence and degree of supernumerary centrosomes in the tissues (spleen, lung, colon, and liver) of 18-month-old Dox-treated *iUVRAG^{FS}* mice as compared to control mice of the same age (Fig. 8a-c). This*

Fig. 8 UVRAG^{FS} promotes centrosome amplification in tissues. **a** Quantification of the level of centrosome amplification in tissues from Dox-treated 18-month-old control and *iUVRAG^{FS}* mice. **b** Quantification of centrosome numbers in tissues from 18-month-old Dox-treated control and *iUVRAG^{FS}* mice. **c** Representative confocal images of centrosomes (immunostained for γ -tubulin in red) in tissues from Dox-treated *iUVRAG^{FS}* mice or control mice. Nuclei were stained with DAPI (blue). **d** Metaphase spread from splenocytes of control and *iUVRAG^{FS}* mice treated with Dox for 2 or 18 months (left). The percentage of cells with abnormal karyotype (aneuploidy) was quantified (right). 50-100 cells were evaluated per mice per experiment.

phenotype was most pronounced in the colon and spleen of *iUVRAG^{FS}* versus control mice (Fig. 8a,b), where high rates of proliferation was observed (Fig. 6 and 7c). In young (2-month-old) mice, despite a visible elevation in the level of centrosome amplification upon *UVRAG^{FS}* expression, no genotype-specific differences were observed with statistical significance in any of the tissues assessed (Fig. S8a,b), indicating that the exacerbated centrosome pathology in older *UVRAG^{FS}* mice genuinely reflects an increase in age-related changes in these organs. Consistent with the consensus that centrosome amplification causes erroneous chromosomal segregation (Fukasawa, 2007), we detected a greater than 3-fold increase in aneuploidy in *UVRAG^{FS}*-expressing splenocytes from both 18-month-old mice and 2-month-old mice to lesser extent (Fig. 8d). In parallel, genome-wide copy number variation (CNV) analysis (Baslan et al., 2012) of tumors isolated from Dox-treated *iUVRAG^{FS}* mice revealed marked variability and heterogeneity with more chromosomal amplifications and deletions than the control (Fig. 8e). Taken together, these data indicate that centrosome abnormalities induced by *UVRAG^{FS}* may play a role in *UVRAG^{FS}*-associated chromosomal aneuploidies that potentially favor tumor formation in *UVRAG^{FS}*-expressing mice. We have also included data on DNA damage relevant to *UVRAG^{FS}* expression in addition to centrosome amplification. We found that the increased proliferation in colons from Dox-treated *iUVRAG^{FS}* mice was always accompanied by a notable increase in the levels of γ -H2AX, a sensitive marker for DNA damage (Rogakou et al., 1998) (Fig. 7c, d). This data, now supplemented in Fig. 7c, provide *in vivo* support for a putative defect in DNA damage repair in cells upon *UVRAG^{FS}* expression, as previously noted *in vitro* (He et al., 2015).

Again, we thank the reviewer for this constructive comment, which strengthens the support for the tumorigenic mechanisms of *UVRAG^{FS}* *in vivo*.

2) The actual tumorigenic power of the FS mutation should be further validated and supported. To do so, the authors are highly recommended to investigate the key findings upon *UVRAG* depletion at least *in vitro*, or in a KO conditional murine model.

Response: We thank the reviewer for this critical comment. Our previous study (He et al., 2015) identified *UVRAG^{FS}* as a *bona fide* dominant-negative mutant of WT *UVRAG*, which interferes with the tumor-suppressing functions of WT *UVRAG* to promote cell proliferation, colony formation, anchorage-independent growth, and xenograft tumor growth *in vitro*. In concordance with *in vitro* studies, we demonstrated in this work *in vivo* that expression of *UVRAG^{FS}* is sufficient to drive spontaneous tumorigenesis through, at least in part, a mechanism that involves aberrant β -catenin stabilization/activation as a result of autophagy suppression. Notably, knockout of *UVRAG* is embryonic

Fig. S7h *UVRAG* knockdown suppresses autophagy and stabilizes β -catenin. SW480 cells were transfected with three different *UVRAG* shRNA. WCL were used for co-IP with anti- β -catenin, followed by IB with the indicated antibodies. **Fig. S7i** Quantitative RT-PCR analysis of indicated gene expression in cells in (h). **Fig. S7j** Colony formation assay of SW480 cells stably expressing control shRNA or *UVRAG* shRNA in (h), after treatment with DMSO or iCRT14.

lethal and a murine model with stage-specific UVRAG knockout is not yet available. To further validate that the tumorigenic power of UVRAG^{FS} is indeed due to UVRAG suppression, rather than other potential effects of UVRAG^{FS}, per Reviewer's suggestion, we have conducted similar experiments in UVRAG-depleted SW480 CRC cells. As expected, we observed that knockdown of endogenous UVRAG resulted in autophagy suppression with an concomitant increase in β -catenin protein levels (Fig. S7h) as well as an increase in the transcriptional output of the Wnt pathway (Fig. S7i). Notably, increased abundance of β -catenin correlated with a marked decrease in β -catenin interaction with the autophagy marker protein LC3-II (Fig. S7h). Moreover, a clear correlation was detected between shRNA knockdown efficiency (shRNA#3>shRNA#2>shRNA#1) and severity of phenotypic changes (shRNA#3>shRNA#2>shRNA#1) (Fig. S7h, i). Consistently, depletion of UVRAG led to increased clonogenic growth in vitro (Fig. S7j), which was effectively ablated by treating cells with iCRT14, an inhibitor of β -catenin signaling that blocks β -catenin-TCF binding (Gonsalves et al., PNAS 2011). Hence, the inhibited clonogenic growth by iCRT14 further supports the regulatory role of β -catenin on tumorigenesis associated with UVRAG suppression and suggests that decreased autophagy is a major contributor to aberrant β -catenin activation in nonchallenged conditions. Interestingly, high expression/activation of β -catenin is clinically implicated in the pathogenesis of DLBCL (Ge et al., Mol. Med. Rep, 2012; Fachel et al., Blood, 2016), a major tumor type that developed in UVRAG^{FS} mice. The significance of the UVRAG^{FS}-autophagy- β -catenin regulatory axis in tumorigenesis is further discussed in the revised manuscript. The new data is now included in Fig. S7h-j and also attached here for the Reviewer's reference.

Minor points:

- The characterization of the conditional Flag/tagged UVRAG^{FS}-luciferase transgene expression was carried out in a very precise and convincing fashion. However, the authors should provide the concept, for which the expression was checked only in spleen and colon. Indeed, part of the following analyses were carried out also in other organs (refer to Fig. 1a, S1a and j).

Response: We detected Dox-inducible UVRAG^{FS} expression in multiple organs. We have now provided new data of induced UVRAG^{FS} expression at the mRNA and protein levels in the lung, liver, as well as in the skeletal muscle and heart in Fig. S1b and S1h. Western blot analysis of Flag-tagged UVRAG^{FS} expression is always included in relevant assays throughout the manuscript whenever the tissues from UVRAG^{FS} mice were used.

-Please, provide a WB analysis for colon in Figure S1c.

Response: The WB analysis for the colon tissue has been provided in Fig. S1c.

- It would be interesting to also investigate on the effects of LDs accumulation observed in livers of starved iUVRAG^{FS} Dox-treated mice. Did the authors observe any lipid-related toxicity in hepatocytes?

Response: Despite the massive accumulation of LDs in livers of Dox-treated UVRAG^{FS} mice, no significant difference in lipotoxicity was observed between two genotypes, as suggested by caspase-3 activation (please refer to Fig. S1j). This may be related to tissue-specific adaptive responses of mice during a short-term fasting.

- Can LD accumulation also reflect an increase in lipids uptake/biosynthesis or lipolysis deregulation beyond autophagic-mediated LDs clearance? Could this accumulation boost liver pathologies such as fatty liver disease and HCC? The authors should address these points in the discussion.

Response: We thank the reviewer for this intriguing comment. Notably, histological analysis of liver samples did not reveal any significant differences between control and UVRAG^{FS} mice under a normal

chow diet, suggesting that expression of $UVRAG^{FS}$ has a minor role in normal lipids uptake/biosynthesis. Starvation switches cellular metabolism from glucose-based to fatty acids-based mitochondrial oxidation. Along this line, we observed a significant accumulation of lipid droplets (LDs) in $UVRAG^{FS}$ mice that was not found in wild-type littermates, suggesting a defect in intracellular lipid breakdown in the liver. Similar observations were also obtained in other autophagy-deficient mice models in the literature (Saito et al., 2019; Settembre et al., 2013). Although liver LDs accumulation upon autophagy deficiency is considered to constitute solid evidence of lipophagy, we cannot formally exclude the possibility that $UVRAG^{FS}$ and/or autophagy inhibition might also dysregulate other processes such as lipolysis beyond lipophagy. In fact, our preliminary results detected higher levels of plasma FFA and glycerol in $UVRAG^{FS}$ mice when compared with controls, which correlated with reduced blood levels of circulating ketone bodies (please refer to the attached images). Since ketone bodies are largely produced in the liver from the oxidation of fatty acids, this result indicates that $UVRAG^{FS}$ -mediated autophagy inhibition may also impair lipid oxidation, as also noted in previous findings (Saito et al., 2019; Settembre et al., 2013). The detailed mechanisms by which $UVRAG^{FS}$ regulates lipid metabolism merits further investigation. Nevertheless, because the scope of this work is not on autophagy in lipid metabolism, we choose to discuss this observation and its potential impact on liver diseases in the Discussion section as suggested by Reviewer, while providing the relevant preliminary results for the Reviewer's reference only.

Fig. 2 Total serum FFA (left), glycerol (middle), and ketones (right) in fed and 48 h-fasted $UVRAG^{FS}$ and control mice.

- “[...] showed massive enlargement and accumulation of lipid droplets (LD) in liver when compared to control mice (Fig. 1c)”. Please, also refer to Fig. S1j.

Response: The revised manuscript has been updated to refer to Fig. S1j.

- In Fig. 1b, make the statistics between Ctr+Dox and iUVRAGFS+Dox, according to what it is shown for panel a.

Response: The relevant statistics have been included in the dot blots of LC3-II/LC3-I and p62/actin in Fig. 1b.

- Negative controls should be included in the Co-immunoprecipitation analyses (e.g. Fig. 1d, 2b, Fig. S2b).

Response: The IgG negative controls have been included in the Co-IP analyses in Fig. 1e (original Fig. 1d), 2b and Fig. S2b.

- The authors should address why increased levels of caspase 1 and of IL-1beta do not match with reduction in the levels of the pro-caspase 1 and -IL-1beta, respectively.

Response: We are grateful for this critical comment. To address this concern, we rigorously re-conducted all of the caspase 1 and IL-1 β Western blot experiments in Fig. 3b, Fig. 4f, Fig. 5h, and Fig. S3g, S7a, and S7b, using different caspase and IL-1 β antibodies: monoclonal anti-caspase-1 (ab108362; Abcam) and polyclonal anti-IL-1 β (AF-4010SP; R&D System). All results

Fig. 3b WB analysis of caspase-1 activation and IL-1 β production in WCLs of BMDM from Dox-treated/untreated $iUVRAG^{FS}$ mice and control mice stimulated with PBS or LPS followed by ATP (LPS+ATP).

showed a consistent and reverse correlation of increased amount of cleaved caspase-1 and mature IL-1 β with decreased amount of pro-caspase-1 and pro-IL-1 β , respectively, and vice versa. We have updated all of the caspase-1 and IL-1 β Western blot data with corresponding re-quantification in the revised manuscript, and also attached one example (Fig. 3b) for the reviewer's reference.

- Fig. 2b, its interpretation in the results section (p. 8) is misleading: "Co-immunoprecipitation analysis showed that LPS treatment resulted in increased interaction...ubiquitination"; indeed, no interaction/ubiquitination is shown in the PBS-treated samples (left side of the image). Rephrase the sentence accordingly.

Response: We have clarified the sentence by rephrasing it to "LPS treatment induced interaction of Beclin1 with MyD88, TRIF, and TRAF6".

-In relation to mtROS production, the authors should also assess $\Delta\Psi_m$ and mitophagy markers.

Response: We thank Reviewer for this comment. Mitochondrial membrane potential ($\Delta\Psi_m$) was measured using TMRM. Consistently, we observed that LPS+ATP treatment induced loss of mitochondrial membrane potential, which was exacerbated in the presence of UVRAG^{FS} expression. This data is included in Fig. S3h and also attached here for the Reviewer's reference. Moreover, Dox-treated iUVRAG^{FS} mice exhibited enhanced recruitment and aggregation of mitophagy marker protein p62, which were either colocalized with or adjacent to TOM20-labelled mitochondria by confocal microscopy. The new data is now included in Fig. S3j.

Fig. S3h Relative mitochondrial membrane potential (Ψ_m) changes in LPS-primed ATP-stimulated iUVRAG^{FS} BMDM in the presence/absence of Dox.

- Could mtROS production be linked to a putative (increased) DNA damage?

Response: Using γ -H2AX as a marker of DNA damage, we found that LPS+ATP treatment induced genomic instability of BMDM, which was further exacerbated by UVRAG^{FS}. This data is now included in Fig. 3b and S3g of revised manuscript.

- Please, analyse chemokines expression also in Fig. 3.

Response: As suggested by the Reviewer, we measured IL-1 β -inducible chemokines CXCL1, CXCL2, and CCL2 expression in the spleens of LPS-stimulated control and iUVRAG^{FS} mice on Dox. We observed a consistent correlation of these chemokine expression with IL-1 β production. The data is included in Fig. S3f of the revised manuscript (also attached here)

Fig. S3f Relative mRNA expression for IL-1 β -inducible chemokines CXCL1, CXCL2, and CCL2 as determined by quantitative RT-PCR in the spleens from LPS-challenged mice of indicated genotypes.

- In relation to Fig. 4d, it is mentioned in p.12 that "enhanced neutrophil infiltration" occurs in the tissue analysed; this is not clear from the image, a higher magnification and/or arrows should be included.

Response: Images of higher magnification have been provided in Fig. 4d with neutrophils marked with arrows. The neutrophil infiltration was confirmed by a pathologist. Cropped zoom-in images of colon section are attached below for the Reviewer's reference.

- Fig. 4l, add the blue/red legend as in Fig. 4i.

Response: The legend has been included in Fig. 4l.

- Together with NLRP3 inhibition by MCC950, authors should also include inhibition of caspase 1 and eventually assess whether one of the two proteins has more relevance in the phenomena observed.

Response: To further justify whether *UVRAG^{FS}*-exacerbated colitis is due to aberrant activation of NLRP3 inflammasome and resultant caspase-1 activation, we comprehensively investigated the activation of other inflammasomes relevant to colitis, including NLRP6, NLRC4, and AIM2. Unlike NLRP3, the complex assembly of NLRP6, NLRC4, and AIM2 inflammasomes by DSS were comparable in colons between control mice and *UVRAG^{FS}*-expressing mice, suggesting a major role for the NLRP3 inflammasome overactivation in *UVRAG^{FS}*-associated colitis in vivo (Fig. 4f-h). Supporting this, *UVRAG^{FS}*-exacerbated colitis was appreciably improved by administration of the NLRP3 inhibitor MCC950 (Fig. 4j-l and Fig. S4j-n). Because activation of multiple inflammasomes all converges on caspase-1 activation, using a caspase-1 inhibitor may not allow differentiation of the specific inflammasome contributing to *UVRAG^{FS}*-related colitis. Therefore, we chose to use a NLRP3 inhibitor in DSS-colitis and supplemented new data of different inflammasomes assembly in *UVRAG^{FS}*-associated increased colitis. The data is now included in Fig. 4f of revised manuscript.

Fig. 4d H&E-stained sections of colon from DSS-treated mice with indicated genotypes on D10 after DSS. Inset magnification highlights inflammation and crypt damage associated with *UVRAG^{FS}* expression. Arrows indicate neutrophil infiltration. Scale bars, 300 μ m.

- Fig. 5a is not clear. Please, provide a higher magnification or crop/zoom in the relevant areas.

Response: Representative macroscopic images with improved resolution of the colon from AOM-DSS-treated control (Ctrl) and *iUVRAG^{FS}* mice are now provided in Fig. 5a of the revised manuscript.

- The accumulation of p62 in Fig. 5h is not convincing.

Response: We have repeated the p62 blot in Fig. 5h and a more representative result was provided, in which expression of *UVRAG^{FS}* led to p62 accumulation along with decreased LC3-II/LC3-I ratio.

- In relation to Fig. 7: did the authors also observed an actual effect of *iUVRAG^{FS}* on the cell cycle?

Response: To confirm whether expression of *UVRAG^{FS}* and resultant β -catenin stabilization has direct effect on cell cycle, we conducted flow cytometry of SW480 colon cancer cells stably expressing vector control or *UVRAG^{FS}*. As expected, concomitant with increased expression of β -catenin target genes *c-Myc* and *Cyclin D1*, we observed accelerated G1/S transition and cell cycle progression (Fig. 7m), an observation that is also consistent with enriched Ki67 staining in *UVRAG^{FS}*-colons. The data is now included in Fig. 7m.

- In Fig. 6a, 7 and Suppl. Fig. 6a is not clear how many mice have been analysed per each group.

Response: The number of mice used in Fig. 6a, Fig. S6a, and Fig. S6b have been updated in the legends. Dot plots were used throughout Fig. 7 to show the number of samples per group and relevant legends were revised.

- The authors should assess whether the EM genes are actually expressed in a β -catenin dependent manner.

Response: We thank the Reviewer for the comment. Despite discrete expression pattern of E-cadherin vs. N-cadherin and Vimentin in colons with UVRAG^{FS} expression, the levels of EMT-related β -catenin was comparable between control mice and UVRAG^{FS}-expressing mice during the onset of short-term (60 days) AOM-DSS treatment. This suggests that EMT induction observed in AOM-DSS-treated UVRAG^{FS} mice is likely independent of the Wnt/ β -catenin pathway. In fact, given the significantly increased production of IL-1 β in AOM-DSS-treated UVRAG^{FS} colons, an IL-1 β -elicited inflammatory response may be a potent inducer of EMT in colitis-associated CRC. The new WB data of β -catenin is now included in Fig. S5c.

- It is recommended to confirm the role of β -catenin by its inhibition at least in vitro.

Response: We thank the Reviewer for this critical comment. To further validate that UVRAG^{FS}-associated β -catenin stabilization and activation is due to UVRAG suppression, rather than other potential effects of UVRAG^{FS} expression, we conducted similar experiments in UVRAG knockdown SW480 cells. As expected, depletion of UVRAG enhanced the abundance of β -catenin and upregulated its downstream target oncogene expression (Fig. S7h, i). Particularly, the degree of β -catenin accumulation correlates with the efficiency of shRNA knockdown (Fig. S7h). Additionally, activation of β -catenin by UVRAG suppression was associated with an obvious increase in clonogenic survival of SW480 cells (Fig. S7j), which was abrogated by treatment of cells with iCRT14, a specific and effective inhibitor of β -catenin-TCF binding (Gonsalves et al., PNAS 2011). Hence, the inhibited clonogenic growth by iCRT14 further supports the regulatory role of β -catenin on tumorigenesis associated with UVRAG suppression and suggests that decreased autophagy is a major contributor to aberrant β -catenin activation in nonchallenged conditions. The new data is now included in Fig. S7h-j. We have also attached the iCRT14 treatment data for the reviewer's reference.

Fig. S7j Colony formation assay of SW480 cells stably expressing control shRNA or three different UVRAG shRNA, after treatment with DMSO or iCRT14 (50 μ M). Data represents the mean \pm SD (n =6). Note shRNA knockdown efficiency is in the order of sh#3>sh#2>sh#1

- Add reference number in the Mat & Met section on Bone marrow-derived macrophages (BMDMs) isolation and culture.

Response: The relevant reference has been included in the Method of revised manuscript.

Finally, we thank the Reviewer for his/her suggestions that have led to a significantly improved manuscript.

Point-by-point Response to Reviewer #2

Reviewer 2 stated that “Overall, the premise for the experiments and the findings are clear. The authors and co-authors should be commended on adding more depth to the understanding, mechanisms, and models in the area of autophagy.”

Response: We are truly appreciative of the reviewer’s enthusiasm and thoughtful suggestions, which were very helpful in revising the manuscript. A detailed response to the Reviewer’s critiques and a description of the new experiments follow:

Critique

1. The interactions of the IP with Beclin1 then IB with Bcl2 and UbK63 are a bit difficult to discern. Is there a way to quantify these findings?

Response: We thank the Reviewer for this suggestion. We have now included the densitometric quantification data of the Bcl-2-associated Beclin1/total Beclin1 ratio and the ubiquitinated (K63)-Beclin1/total Beclin1 ratio in Fig. 2b (also attached here for Reviewer’s reference) and Fig. S2b. Consistent results were obtained in repeated experiments such that Beclin1 interaction with its negative regulator Bcl-2 was significantly increased in LPS-stimulated BMDM and tissues expressing UVRAG^{FS}, concomitant with decreased levels of Beclin1 K63 ubiquitination. The new data is included in Fig. 2b and Fig. S2b of revised manuscript.

From Fig. 2b Densitometric quantification of the Bcl-2-associated Beclin1/total Beclin1 and the ubiquitinated (K63)-Beclin1/total Beclin1 ratios in LPS-stimulated BMDM. Data represents the mean \pm SD (n = 4 independent experiments).

2. The data in Figure 7k, regarding p62 are not convincing.

Response: We have rigorously re-conducted the p62 Western blot in Fig. 7k and more representative data has been provided (also attached here for reference). We also supplemented the corresponding densitometric quantification of the p62/Actin ratio to demonstrate dose-dependent decrease of autophagy in UVRAG^{FS}-expressing SW480 cells.

from Fig. 7k: UVRAG^{FS} inhibits autophagy-mediated turnover of p62 in a dose dependent manner.

3. Since the Rosa26 transgenic expression is global, why were only BMDM studied?

Response: We would like to highlight that: **1)** for LPS-induced autophagy activation and its suppression by UVRAG^{FS}, different tissues, including liver, colon, and lung were examined, in addition to BMDM, and consistent results were obtained (Fig. 2a, 2c, and S2c); **2)** for the molecular mechanism related to K63 ubiquitination of Beclin1, both BMDM and liver tissues were used to corroborate the findings (Fig. 2b and S2b); **3)** for autophagy-regulated inflammatory responses, pro-inflammatory cytokine production was evaluated in serum, spleen, as well as BMDM (Fig. 3 and S3); **4)** finally, for autophagy-related inflammasome activation in LPS-sepsis, BMDM is mainly used because macrophages are the major sources of the pro-inflammatory cytokines responsible for the development of LPS-induced septic shock. Additionally, ATP-driven activation of caspase-1 in LPS-primed macrophages is an established model for inflammasome-mediated caspase-1 activation in vitro

(Sutterwala et al., 2006). Of note, in DSS-induced colitis, colon tissues were also used to study intestinal inflammation regulated by UVRAG^{FS} in vivo.

4. Was there an attempt to create an inducible UVRAGFS murine model with expression only in the colon, or in the small bowel and proximal colon?

Response: Thank Reviewer for this thoughtful suggestion. Indeed, the Dox-inducible and colon-specific UVRAG^{FS} Tg mice is under our active development by crossing the TRE-Flag-UVRAG^{FS} mice with the transactivator mice that express rtTA under the Fabp1 promoter (intestinal tissue-specific). We believe that this model, once developed, will provide important insights into the tissue-specific role of UVRAG^{FS} in inflammation and tumorigenesis. Nonetheless, in the current work, we mainly focused on the systematic impact of UVRAG^{FS} expression in autophagy, inflammation, and tumorigenesis.

5. Were other models of enteritis or colitis such as in Samp-Yit or Winnie mice explored?

Response: We thank the Reviewer for this thoughtful comment. To understand the effect of autophagy suppression on inflammasome activation, DSS-induced colitis murine model was mainly used because it is a well-established model for the study of intestinal inflammation, particularly those mediated by the inflammasome response (Bauer et al., 2010). In addition to NLRP3 inflammasome, we also observed increased complex assembly of the NLRP6, NLRC4, and AIM2 inflammasomes during the onset of DSS treatment in mice of both genotypes (Fig. 4f). However, expression of UVRAG^{FS} only, caused increased NLRP3 inflammasome activation and resultant IL-1 β overproduction, but showed minimal effects on other inflammasomes' assembly (Fig. 4f), suggesting a specific role of autophagy suppression in aggravating NLRP3-mediated inflammation. Consistent with this finding, UVRAG^{FS}-associated colitis could be alleviated by MCC950, a specific inhibitor of NLRP3 inflammasome (Fig. 4j-l). In addition to chemical-induced colitis, it would be very interesting to also investigate the impact of UVRAG^{FS} on the spontaneous chronic colitis mouse model such as Winnie mice, as suggested by the Reviewer. Importantly, the Winnie mouse has missense mutations in the mucin Muc2 gene and thus defects in goblet cells (McGuckin et al., 2010) and loss of function of autophagy also perturbs goblet cell functions (Patel et al., 2013). Thus, it is speculated that expression of UVRAG^{FS} and by extension autophagy suppression might exacerbate colonic inflammation in Winnie mice likely by multifold mechanisms. Nevertheless, inclusion of additional colitis models might be overwhelming in size and out of scope of current work at this stage. Thus, we will save this idea for future studies.

6. Were other acute or chronic chemokine/cytokine expression noted in the KC, MIP2, CXCL1 families?

Response: Thank Reviewer for this comment. In addition to the previously analyzed CXCL1 (also known as KC), CXCL2 (also known as MIP2), and CCL2 (also known as MCP-1), we further evaluated other chemotactic factors in the CXC and CC chemokine subfamily inducible by IL-1 β , including CXCL3, CCL3 (also known as MIP1 α), and CXCL10. Consistently, we observed a similar pattern of induction of these chemokine expression. These results are consistent with enhanced influx of neutrophils in DSS-fed colons upon UVRAG^{FS} expression. The new data has been included in Fig. S4d and also attached here for the reviewer's reference.

Fig. S4d Relative mRNA expression of the neutrophil chemokines CXCL1, CXCL2, CXCL3, CCL2, CCL3, and CXCL10 as determined by quantitative RT-PCR in the colons from mice of indicated genotypes.

7. In Figure 5e, was only a single mouse used? Is there quantification of the E-cadherin and N-cadherin?

Response: *The IHC staining of indicated proteins is from one animal that is representative of 5-12 animals in each group (as noted in figure legend). The quantification of EMT-related proteins is included in Fig. S5c from repeated experiments.*

8. What are the thoughts regarding the localization and types of malignancies that spontaneously developed (DLBCL)?

Response: *In our aged cohorts, Dox-treated iUVRAG^{FS} mice had mainly developed spontaneous B cell lymphomas, particularly DLBCL, and exhibited hyperplastic response in colons. We have demonstrated that the tumorigenic power of UVRAG^{FS} is associated with, at least in part, increased abundance of β -catenin oncoprotein as a result of accelerated age-related autophagy suppression. Given the essential role of the Wnt/ β -catenin pathway in intestinal homeostasis and tumorigenesis (Clevers, 2004; Fodde and Brabletz, 2007), it might not be surprising to detect colonic hyperplastic changes in UVRAG^{FS}-expressing mice. Indeed, global expression of the UVRAG^{FS} mutant is detected in human colon cancers, especially those with microsatellite instability (He et al., 2015). Although UVRAG mutation and/or downregulation has not yet been reported in DLBCL, overexpression and aberrant activation of β -catenin has been found to be strongly correlated to the pathogenesis of DLBCL and represents a new target for DLBCL lymphomagenesis (Bognar et al., 2016; Ge et al., 2012; Walker et al., 2015). In this regard, our iUVRAG^{FS} model is valuable not only for understanding tumorigenesis associated with UVRAG suppression, but only for studying the mechanisms of DLBCL development. We have included these thoughts in the discussion section of the revised manuscript.*

9. Is there an area in the supplementary materials where the entire western blots are available rather than cropped versions?

Response: *The uncropped western blots presented in the figures are now included in Fig.S9.*

Again, we thank the Reviewer for his/her thoughtful suggestions that have led to a much-improved manuscript.

Point-by-point Response to Reviewer #3

Reviewer 3 commented that “This is a very interesting paper; the experiments are well performed and the results are convincing. However, there are several loose ends that need to be tightened prior to publication.”

Response: We are truly appreciative of the reviewer’s enthusiasm and encouraging comments, which were very helpful in revising the manuscript. A detailed response to the Reviewer’s critiques and a description of the new experiments follow:

Critique

1. In the LPS endotoxemia model, caspase-11 is the key inflammasome involved in IL-1b and IL-18 production. The authors need to show caspase-11 levels and activation. They presume based on the BMDM ex vivo results that NLRP3 is involved in mediating the hyper-inflammatory response in this endotoxemia model. This needs to be demonstrated. The authors can cross their mice to NLRP3-deficient mice, or as they did in the DSS colitis model, they can use the NLRP3 inhibitor MCC-950.

Response: We thank the Reviewer for this important suggestion. As noted, unlike caspase-1 which is activated by canonical inflammasomes, caspase-11 can be activated by direct LPS binding (Shi et al., 2014) and thus plays a critical role in endotoxic shock. To further clarify this in our model, we evaluated the levels and activation of caspase-11. We found that despite a notable increase in caspase-1 activation (indicated by cleaved caspase-1) in LPS+ATP-treated *iUVRAG^{FS}* BMDM on Dox, activation of the inflammasome-independent inflammatory caspase, caspase-11, by LPS was similar in *UVRAG^{FS}*-expressing BMDM relative to its activation in the control cells (Fig. 3b). Thus, caspase-1 overactivation largely contributes to excessive inflammation in Dox-treated *iUVRAG^{FS}* mice after LPS stimulation. The new data has been included in Fig. 3b, and also attached here for the reviewer’s reference.

To further demonstrate the specific role of the NLRP3 inflammasome in *UVRAG^{FS}*-associated inflammatory response in septic shock, we used MCC950, a small-molecule inhibitor of the NLRP3 inflammasome (Coll et al., 2015), to treat mice under septic shock. As expected, administration of MCC950 decreased the levels of IL-1 β in serum (Fig. 3f) and improved the survival rate of *UVRAG^{FS}* mice treated with LPS (Fig. 3e), suggesting that aberrant activation of NLRP3 inflammasome directly contributes to excessive inflammation associated with *UVRAG^{FS}* expression during sepsis. The new data has been included in Fig. 3e and 3f, and also attached here for the reviewer’s

Fig. 3b

Fig. 3e

Fig. 3f

Fig. 3b Co-IP of ASC and NLRP3 with pro-caspase-1 and WB analysis of activation of caspase-1, caspase-11, and IL-1 β production in WCLs and/or supernatants (Sup.) of BMDM from Dox-treated/untreated *iUVRAG^{FS}* mice and control mice stimulated with PBS or LPS+ATP. **Fig. 3e** Survival of Dox-treated *iUVRAG^{FS}* mice in response to MCC950 therapy in LPS-induced sepsis. **Fig. 3f** ELISA of serum IL-1 β of Dox-treated control and *iUVRAG^{FS}* mice in response to MCC950 therapy in LPS-induced sepsis.

reference.

2. It has been recently reported that not only NLRP3 but also AIM2 inflammasome is regulated by autophagy (and not mitophagy) through control of ASC levels. Therefore, ASC levels should be shown in the different models. Also, it will be interesting to examine the response of BMDM to poly(dA:dT) to assess the AIM2 inflammasome.

Response: We thank the Reviewer for this suggestion. Levels of the ASC adaptor protein of inflammasomes are included in all inflammasome-relevant dataset throughout the manuscript (Fig. 3b, 4f, 5h, S3g, S7a, S7b). No significant difference was detected between Dox-treated control and *iUVRAG^{FS}* mice during the onset of LPS-sepsis, DSS-colitis, and spontaneous tumorigenesis in repeated experiments. Moreover, we found that expression of *UVRAG^{FS}* did not alter complex formation of the AIM2 inflammasome, nor the NLRP6 and NLRC4 inflammasomes, but specifically induced increased NLRP3 assembly/activation and perpetuated inflammation that can be alleviated by MCC950 (Fig. 4f). Thus, although autophagy suppression could decrease the turnover of inflammasomes (e.g. AIM2, ASC) in the lysosome as documented in the literature (Shi et al., 2012), this degradative ability might be compensated by inducers of these inflammasomes that cause increased activation of autophagy in a context-dependent manner. The complex interplay between autophagy and the inflammasome machineries merits further investigation.

Per the Reviewer's suggestion, we have examined the AIM2 inflammasome activation in response to cytoplasmic DNA in Dox-treated *iUVRAG^{FS}* BMDM. We observed that *UVRAG^{FS}*-expressing BMDM responded normally to poly(dA:dT), when compared to control BMDM, in terms of caspase-1 activation and IL-1 β production. Consistent with that observation, assembly of the AIM2-ASC-procaspase-1 complex was comparable between Dox-treated *iUVRAG^{FS}* BMDM and wild-type cells. Hence, in our context, *UVRAG^{FS}*-mediated autophagy suppression does not appear to have a dominant role in the regulation of AIM2 inflammasome even in response to cytosolic dsDNA. We have included this data in Fig. S3k (also attached here for the reviewer's reference) and updated the manuscript accordingly.

3. In the DSS colitis model, IL-18 production is essential in tissue repair. Further, different inflammasomes are activated in this model including NLRP6, NLRC4, AIM2 (all 3 in the epithelium) and NLRP3 (in myeloid cells). It will be important to show levels and activation of these inflammasomes and levels of IL-18.

Response: We thank the Reviewer for this thoughtful suggestion. We have conducted the experiments suggested by the Reviewer and found that DSS-treated *UVRAG^{FS}* mice colons exhibited enhanced cleavage of pro-IL-18, as seen with that of pro-IL-1 β , into its biologically active form IL-18, which was associated with increased NLRP3 inflammasome assembly and caspase-1 activation (Fig. 4f-h; also

From Fig. 4f. Inflammasome assembly, caspase-1 cleavage, and the production of IL-1 β and IL-18 in control and *UVRAG^{FS}* mice during DSS-colitis.

Fig. S3k Co-IP of ASC and AIM2 with pro-caspase-1 and WB analysis of activation of caspase-1 and IL-1 β production in WCLs of BMDM from Dox-treated/untreated *iUVRAG^{FS}* mice and control mice transfected with Poly(dA:dT).

attached here for reference). However, formation of other colitis-relevant inflammasomes such as AIM2, NLRP6, and NLRC4 by DSS was comparable in colons between control mice and UVRAG^{FS}-expressing mice in repeated experiments (Fig. 4f), highlighting a critical role of NLRP3 inflammasome in UVRAG^{FS}-associated colitis. As seen with IL-1 β production, amounts of IL-18, but not IL-6 and TNF- α , produced by colonic tissue were significantly increased in DSS-fed UVRAG^{FS}-expressing mice versus DSS-fed control mice (Fig. 4i, right panel). We have included the new data of IL-18 production as well as colitis-associated inflammasome activation in Fig. 4f-i of the revised manuscript.

4. What is the autophagy target in B cell lymphoma? And lung adenoma? The authors should discuss these findings.

Response: We thank the Reviewer for this suggestion. In our aged cohorts, Dox-treated iUVRAG^{FS} mice had mainly developed spontaneous B cell lymphomas, particularly DLBCL, some lung adenoma, and exhibited a hyperplastic response in colons. We have demonstrated that the tumorigenic power of UVRAG^{FS} is mechanistically associated with, at least in part, increased abundance of β -catenin oncoprotein as a result of increased age-related decline/suppression of autophagy. In fact, our data are consistent with the previous *in vitro* finding that the autophagy pathway is important in β -catenin degradation (Sukhdeo et al., 2012). Given the essential role of the Wnt/ β -catenin pathway in intestinal homeostasis and tumorigenesis (Clevers, 2004; Fodde and Brabletz, 2007), it is not surprising to detect colonic hyperplasia in UVRAG^{FS} mice. Indeed, global expression of the UVRAG^{FS} mutant is detected in human colon cancers, particularly those with microsatellite instability (He et al., 2015). Although UVRAG mutation and/or downregulation has not yet been reported in DLBCL, overexpression and aberrant activation of β -catenin has been found to be strongly correlated to the pathogenesis of DLBCL and represents a new target for DLBCL lymphomagenesis (Bognar et al., 2016; Ge et al., 2012; Walker et al., 2015). It also has substantial impacts on non-small lung cancer (NSCLC) tumorigenesis and prognosis (Jin et al., 2017; Nakayama et al., 2014) and thus warrants further exploration. We have included these thoughts in the discussion section of the revised manuscript.

5. In the colon hyperplastic response, while the authors excluded NLRP3 contribution, they haven't excluded a role of other inflammasomes (that could be hyperactivated due to ASC accumulation), and of IL-18!! This is important as this cytokine is involved in IEC proliferation.

Response: We have now included the WB analysis of IL-18 cleavage, in parallel with IL-1 β , as well as other inflammasome protein expression in the study of spontaneous tumorigenesis. Consistently, as seen with IL-1 β , no notable IL-18 production and caspase-1 activation was detected in both young (2-month-old) and older (18-month-old) mice in either genotype (Fig. S7a). Likewise, no changes were observed in the expression of inflammasomes-related proteins (i.e. ASC, NLRP3, NLRP6, AIM2, and NLRC4) in these mice (Fig. S7a). Similar results were obtained in splenocytes that gave rise to increased spontaneous malignancies (Fig. S7b). It is important to note that irrespective of activation of multiple inflammasomes, they all converge on the cleavage/activation of caspase-1. In our model of spontaneous tumorigenesis, no caspase-1 activation had ever been detected in both young and aged mice. This suggests that inflammasome activation is not a major participant in promoting spontaneous tumorigenesis associated with UVRAG^{FS} expression. The new data is now included in Fig. S7a, S7b of the revised manuscript.

Finally, we thank the Reviewer for his/her thoughtful suggestions that have led to a much-improved manuscript.

Reference:

- Baslan, T., Kendall, J., Rodgers, L., Cox, H., Riggs, M., Stepansky, A., Troge, J., Ravi, K., Esposito, D., Lakshmi, B., *et al.* (2012). Genome-wide copy number analysis of single cells. *Nat Protoc* 7, 1024-1041.
- Bauer, C., Duewell, P., Mayer, C., Lehr, H.A., Fitzgerald, K.A., Dauer, M., Tschopp, J., Endres, S., Latz, E., and Schnurr, M. (2010). Colitis induced in mice with dextran sulfate sodium (DSS) is mediated by the NLRP3 inflammasome. *Gut* 59, 1192-1199.
- Bognar, M.K., Vincendeau, M., Erdmann, T., Seeholzer, T., Grau, M., Linnemann, J.R., Ruland, J., Scheel, C.H., Lenz, P., Ott, G., *et al.* (2016). Oncogenic CARMA1 couples NF-kappaB and beta-catenin signaling in diffuse large B-cell lymphomas. *Oncogene* 35, 4269-4281.
- Clevers, H. (2004). Wnt signaling: Ig-norrin the dogma. *Current biology : CB* 14, R436-437.
- Coll, R.C., Robertson, A.A., Chae, J.J., Higgins, S.C., Munoz-Planillo, R., Inserra, M.C., Vetter, I., Dungan, L.S., Monks, B.G., Stutz, A., *et al.* (2015). A small-molecule inhibitor of the NLRP3 inflammasome for the treatment of inflammatory diseases. *Nat Med* 21, 248-255.
- Fodde, R., and Brabletz, T. (2007). Wnt/beta-catenin signaling in cancer stemness and malignant behavior. *Curr Opin Cell Biol* 19, 150-158.
- Fukasawa, K. (2007). Oncogenes and tumour suppressors take on centrosomes. *Nat Rev Cancer* 7, 911-924.
- Ge, X., Lv, X., Feng, L., Liu, X., and Wang, X. (2012). High expression and nuclear localization of beta-catenin in diffuse large B-cell lymphoma. *Molecular medicine reports* 5, 1433-1437.
- He, S., Zhao, Z., Yang, Y., O'Connell, D., Zhang, X., Oh, S., Ma, B., Lee, J.H., Zhang, T., Varghese, B., *et al.* (2015). Truncating mutation in the autophagy gene UVRAG confers oncogenic properties and chemosensitivity in colorectal cancers. *Nature communications* 6, 7839.
- Jin, J., Zhan, P., Katoh, M., Kobayashi, S.S., Phan, K., Qian, H., Li, H., Wang, X., Wang, X., Song, Y., *et al.* (2017). Prognostic significance of beta-catenin expression in patients with non-small cell lung cancer: a meta-analysis. *Transl Lung Cancer Res* 6, 97-108.
- McGuckin, M.A., Eri, R.D., Das, I., Lourie, R., and Florin, T.H. (2010). ER stress and the unfolded protein response in intestinal inflammation. *Am J Physiol Gastrointest Liver Physiol* 298, G820-832.
- Nakayama, S., Sng, N., Carretero, J., Welner, R., Hayashi, Y., Yamamoto, M., Tan, A.J., Yamaguchi, N., Yasuda, H., Li, D., *et al.* (2014). beta-catenin contributes to lung tumor development induced by EGFR mutations. *Cancer Res* 74, 5891-5902.
- Patel, K.K., Miyoshi, H., Beatty, W.L., Head, R.D., Malvin, N.P., Cadwell, K., Guan, J.L., Saitoh, T., Akira, S., Seglen, P.O., *et al.* (2013). Autophagy proteins control goblet cell function by potentiating reactive oxygen species production. *The EMBO journal* 32, 3130-3144.
- Rogakou, E.P., Pilch, D.R., Orr, A.H., Ivanova, V.S., and Bonner, W.M. (1998). DNA double-stranded breaks induce histone H2AX phosphorylation on serine 139. *The Journal of biological chemistry* 273, 5858-5868.
- Saito, T., Kuma, A., Sugiura, Y., Ichimura, Y., Obata, M., Kitamura, H., Okuda, S., Lee, H.C., Ikeda, K., Kanegae, Y., *et al.* (2019). Autophagy regulates lipid metabolism through selective turnover of NCoR1. *Nature communications* 10, 1567.
- Settembre, C., De Cegli, R., Mansueto, G., Saha, P.K., Vetrini, F., Visvikis, O., Huynh, T., Carissimo, A., Palmer, D., Klisch, T.J., *et al.* (2013). TFEB controls cellular lipid metabolism through a starvation-induced autoregulatory loop. *Nat Cell Biol* 15, 647-658.
- Shi, C.S., Shenderov, K., Huang, N.N., Kabat, J., Abu-Asab, M., Fitzgerald, K.A., Sher, A., and Kehrl, J.H. (2012). Activation of autophagy by inflammatory signals limits IL-1beta production by targeting ubiquitinated inflammasomes for destruction. *Nat Immunol* 13, 255-263.
- Shi, J., Zhao, Y., Wang, Y., Gao, W., Ding, J., Li, P., Hu, L., and Shao, F. (2014). Inflammatory caspases are innate immune receptors for intracellular LPS. *Nature* 514, 187-192.
- Sukhdeo, K., Mani, M., Hideshima, T., Takada, K., Pena-Cruz, V., Mendez, G., Ito, S., Anderson, K.C., and Carrasco, D.R. (2012). beta-catenin is dynamically stored and cleared in multiple myeloma by the proteasome-aggresome-autophagosomal-lysosome pathway. *Leukemia* 26, 1116-1119.
- Sutterwala, F.S., Ogura, Y., Szczepanik, M., Lara-Tejero, M., Lichtenberger, G.S., Grant, E.P., Bertin, J., Coyle, A.J., Galan, J.E., Askenase, P.W., *et al.* (2006). Critical role for NALP3/CIAS1/Cryopyrin in innate and adaptive immunity through its regulation of caspase-1. *Immunity* 24, 317-327.
- Walker, M.P., Stopford, C.M., Cederlund, M., Fang, F., Jahn, C., Rabinowitz, A.D., Goldfarb, D., Graham, D.M., Yan, F., Deal, A.M., *et al.* (2015). FOXP1 potentiates Wnt/beta-catenin signaling in diffuse large B cell lymphoma. *Science signaling* 8, ra12.

REVIEWERS' COMMENTS:

Reviewer #1 (Remarks to the Author):

The authors satisfactorily addressed all my comments. In order to further improve the quality and overall uniformity of the manuscript, I dare to suggest some minor additional changes:

1. mention in the abstract the very interesting effect of UVRAGFS on centrosome homeostasis in vivo, which greatly supports its tumorigenic role;
2. provide a full image for caspase-1 (pro- and cleaved) in Figures 3b, S3g, S7a and S7b, as the authors did in Figures 4f and 5h.
3. normalize the mitochondrial membrane potential provided in Fig. S3h on the total amount of mitochondria (e.g. by means of MTG or similar).

Reviewer #2 (Remarks to the Author):

This is a revised manuscript from Quach et al, examining the impact of a frameshift mutation in the autophagy gene UVRAG. The initial submission was well-received conceptually, however, several details were difficult to visualize or to comprehend thus rendering the manuscript a bit challenging to follow and to support. The majority of my queries have now been satisfied. the authors are to be congratulated for this body of work.

Minor

1. The result concerning Figure 5e, h and p62 is difficult to substantiate, regarding turnover.
2. In Figure 7g, again, not clear regarding p62 turnover.
3. A few typographical errors: line 267 should have 'levels' instead of 'levers'; the label for figure 5h 'homeogenate' rather than 'homogenate'.
4. Supplementary figure 9 in several cases still does not demonstrate the entire image, but for the most part, the area from where the cropping has been done is in context.

Reviewer #3 (Remarks to the Author):

The authors addressed all of my comments. The paper is substantially improved and acceptable for publication.

RESPONSE TO REVIEWERS

We thank all Reviewers to finally approve our manuscript for publication. We have revised our manuscript to comply with the journal's format requirements and to maximize the accessibility and the impact of work as instructed. Track changes were used to make changes in the maintext.

Reviewer #1 (Remarks to the Author):

The authors satisfactorily addressed all my comments. In order to further improve the quality and overall uniformity of the manuscript, I dare to suggest some minor additional changes:

1. mention in the abstract the very interesting effect of UVRAGFS on centrosome homeostasis in vivo, which greatly supports its tumorigenic role;
Response: the abstract has been updated as suggested.
2. provide a full image for caspase-1 (pro- and cleaved) in Figures 3b, S3g, S7a and S7b, as the authors did in Figures 4f and 5h.
Response: full images for caspase-1 in Figures 3b, S3g, S7a and S7b have been provided.
3. normalize the mitochondrial membrane potential provided in Fig. S3h on the total amount of mitochondria (e.g. by means of MTG or similar).
Response: the mitochondrial membrane potential provided in Fig. S3h is normalized on the total amount of mitochondria by means of MTG.

Reviewer #2 (Remarks to the Author):

This is a revised manuscript from Quach et al, examining the impact of a frameshift mutation in the autophagy gene UVRAG. The initial submission was well-received conceptually, however, several details were difficult to visualize or to comprehend thus rendering the manuscript a bit challenging to follow and to support. The majority of my queries have now been satisfied. the authors are to be congratulated for this body of work.

Minor

1. The result concerning Figure 5e, h and p62 is difficult to substantiate, regarding turnover.
Response: new p62 images in Figure 5e and h have been provided
2. In Figure 7g, again, not clear regarding p62 turnover.
Response: new p62 image in Figure 7g has been provided
3. A few typographical errors: line 267 should have 'levels' instead of 'levers'; the label for figure 5h 'homeogenate' rather than 'homogenate'.
Response: Typographical errors have been corrected.
4. Supplementary figure 9 in several cases still does not demonstrate the entire image, but for the most part, the area from where the cropping has been done is in context.
Response: new Supplementary Figure 9 has been provided including the entire images.

Reviewer #3 (Remarks to the Author):

The authors addressed all of my comments. The paper is substantially improved and acceptable for publication.

Finally, we thank all Reviewers for a much-improved manuscript and a granted publication in principle.

Thank you!